# Dynamic rhenium dopant boosts ruthenium oxide for durable oxygen evolution

Huanyu Jin [1,2,8], Xinyan Liu[3,8], Pengfei An[4], Cheng Tang[1], Huimin Yu[5], Qinghua Zhang[6], Hong-Jie Peng [3], Lin Gu[6], Yao Zheng [1], Taeseup Song [7], Kenneth Davey [1], Ungyu Paik [7], Juncai Dong [4] ✉ & Shi-Zhang Qiao [1] ✉

Heteroatom-doping is a practical means to boost $RuO_2$ for acidic oxygen evolution reaction (OER). However, a major drawback is conventional dopants have static electron redistribution. Here, we report that Re dopants in $Re_{0.06}Ru_{0.94}O_2$ undergo a dynamic electron accepting-donating that adaptively boosts activity and stability, which is different from conventional dopants with static dopant electron redistribution. We show Re dopants during OER, (1) accept electrons at the on-site potential to activate Ru site, and (2) donate electrons back at large overpotential and prevent Ru dissolution. We confirm via in situ characterizations and first-principle computation that the dynamic electron-interaction between Re and Ru facilitates the adsorbate evolution mechanism and lowers adsorption energies for oxygen intermediates to boost activity and stability of $Re_{0.06}Ru_{0.94}O_2$. We demonstrate a high mass activity of 500 A $g_{cata.}^{-1}$ (7811 A $g_{Re-Ru}^{-1}$) and a high stability number of S-number = $4.0 \times 10^6$ $n_{oxygen}$ $n_{Ru}^{-1}$ to outperform most electrocatalysts. We conclude that dynamic dopants can be used to boost activity and stability of active sites and therefore guide the design of adaptive electrocatalysts for clean energy conversions.

Proton exchange membrane (PEM) water electrolyzers are practically promising for green hydrogen production because of the high current density, energy efficiency, and safety[1-4]. However, sustainable application is hindered by high price and low reserve of iridium (Ir), the anode material for oxygen evolution reaction (OER)[4-6]. For example, the demand for Ir is nine times greater than current global production for all planned PEM electrolyzers for 2030 in the *Net Zero Emissions Scenario*[1]. Therefore, efficient and Ir-free OER electrocatalysts are important for continuing competitiveness of PEM water electrolyzers[7].

In 1987 researchers reported that rutile-phase ruthenium oxide ($RuO_2$) has significant practical potential to replace $IrO_2$ for acidic OER

because of excellent activity[8]. However, achieving high activity and stability with this catalyst is practically difficult because of the instability of surface Ru sites in $RuO_2$[9-11]. Ru active sites are damaged in two ways, (1) formation of lattice-oxygen vacancies via lattice-oxygen-mediated mechanism (LOM)[12-14], which leads to instability of the oxygen anion[15], and (2) over-oxidation of Ru atoms to soluble $RuO_4$ species at the high overpotential that leads to demetallation of the active sites[16]. These two ways are correlated with each other, leading to the rapid degradation of Ru active sites. In recent years, research has focused on boosting structural stability of $RuO_2$ or Ru-based electrocatalysts via hybridization with $IrO_2$[17-24]. However, maintaining

[1]School of Chemical Engineering and Advanced Materials, The University of Adelaide, Adelaide, SA 5005, Australia. [2]Institute for Sustainability, Energy and Resources, The University of Adelaide, Adelaide, SA 5005, Australia. [3]Institute of Fundamental and Frontier Sciences, University of Electronic Science and Technology of China, Chengdu 611731 Sichuan, China. [4]Beijing Synchrotron Radiation Facility, Institute of High Energy Physics, Chinese Academy of Sciences, Beijing 100049, China. [5]Future Industries Institute, University of South Australia, Mawson Lakes Campus, Adelaide, SA 5095, Australia. [6]Beijing National Laboratory for Condensed Matter Physics, Institute of Physics, Chinese Academy of Sciences, Beijing 100190, China. [7]Department of Energy Engineering, Hanyang University, Seoul 04763, Republic of Korea. [8]These authors contributed equally: Huanyu Jin, Xinyan Liu. ✉e-mail: dongjc@ihep.ac.cn; s.qiao@adelaide.edu.au

sufficient activity is practically difficult because the Ir dopants limit the flexibility of Ru redox[25,26].

Theoretically, the over-oxidation of Ru site is hindered thermodynamically if the formation energy of oxygen vacancies is greater than that for redox $H_2O/O_2$[27]. For example, it was reported that stability and activity of $RuO_2$ nanoparticles can be boosted by reinforcing Ru−O bonding via surface heat treatment[10]. In recent years, foreign element-doping was developed to reinforce Ru−O bonding via enlarging the localized gap between O $2p$ band centres and Fermi level[3,28–30]. For example, co-doping of W and Er tuned electronic structure of $RuO_2$ via charge redistribution that significantly reduced Ru dissolution and lowered adsorption energies for oxygen intermediates[31]. However, most reported dopants on $RuO_2$ have fixed interaction with the Ru site, resulting in limited success in boosting activity or stability[32–35]. Even though conventional heteroatom doping strengthens the lattice oxygen in $RuO_2$ at small overpotential via electronic structural redistribution, the stability is not sufficient for practical application because of demetallation of the modified-Ru site at large overpotentials[36]. Therefore, dopants that can tune OER performance via dynamic electron distribution under differing potentials are practically attractive. Reported findings have demonstrated incorporation of high-valence metals, such as W and Ta to stabilize low-charge Ir/Ru[31,37]. Doping with Re, an element with multiple oxidation states between −3 and +7, into $RuO_2$, it is possible to build adaptive active sites with dynamic electron transfer. In addition, rhenium oxide is an acidic oxide with strong metal−oxygen bonds. Therefore, Re dopants stabilize the lattice oxygen and low-charge metal active sites concurrently.

Here we report a Re-doped rutile $RuO_2$ ($Re_{0.06}Ru_{0.94}O_2$) electrocatalyst that exhibits excellent performance for acidic OER. We show that Re dopants in $Re_{0.06}Ru_{0.94}O_2$ undergo a dynamic electron accepting-donating that adaptively boosts activity and stability, and that this is different from conventional dopants with static dopant electron redistribution. We use operando X-ray absorption spectroscopy (XAS) to confirm that the Re dopants gain electrons from Ru site at the on-site potential to activate OER, and donate electrons back at large overpotential to prevent Ru dissolution. We demonstrate that activity and stability of $Re_{0.06}Ru_{0.94}O_2$ are therefore significantly boosted. We confirm using judiciously combined in situ characterizations and density functional theoretical (DFT) computation that the Re doping reduces adsorption energies for oxygen intermediates, and reinforces the Ru−O bonding in the adsorbate evolution mechanism (AEM) pathway. We conclude that dynamic dopants can be used to boost activity and stability of active sites and therefore guide practical design of adaptive electrocatalysts.

## Results

### Electrocatalytic performance for $Re_{0.06}Ru_{0.94}O_2$

The Re was doped in $RuO_2$ via a molten-salt method, and the chemical formula for $Re_{0.06}Ru_{0.94}O_2$ was determined by inductively coupled plasma mass spectrometry (ICP-MS) (Supplementary Fig. 1). The pristine $RuO_2$ and $Re_{0.06}Ru_{0.94}O_2$ were analyzed by different characterizations (Supplementary Figs. 2–6). OER performances for $Re_{0.06}Ru_{0.94}O_2$ and other control samples were determined in $O_2$-saturated 0.1 M $HClO_4$ electrolyte. Figure 1a, b presents the linear sweep voltammetry (LSV) curves and corresponding Tafel slopes for $Re_{0.06}Ru_{0.94}O_2$, $RuO_2$, and commercial $RuO_2$ (denoted C-$RuO_2$). $Re_{0.06}Ru_{0.94}O_2$ exhibited an overpotential of 190 mV at current density 10 mA cm$^{-2}$ ($\eta_{10}$) with a Tafel slope 45.5 mV dec$^{-1}$, outperforming $RuO_2$ with 258 mV and 50.3 mV dec$^{-1}$ and, C-$RuO_2$, 388 mV and 76.4 mV dec$^{-1}$, respectively[38]. LSV curves without i-R compensation are presented in Supplementary Fig. 7a as a reference. The mass activity for $Re_{0.06}Ru_{0.94}O_2$ is 500 A g$^{-1}$ at overpotential 272 mV, which is greater than $RuO_2$ of 156 A g$^{-1}$ at 272 mV and, C-$RuO_2$ of 18 A g$^{-1}$ at 290 mV, respectively (Supplementary Fig. 7b). In addition, Re-correlated Ru sites (Ru active sites linked to Re atoms) exhibit a mass activity of 7811 A g$^{-1}$ (Fig. 1c), which is greater than most acidic OER catalysts. The

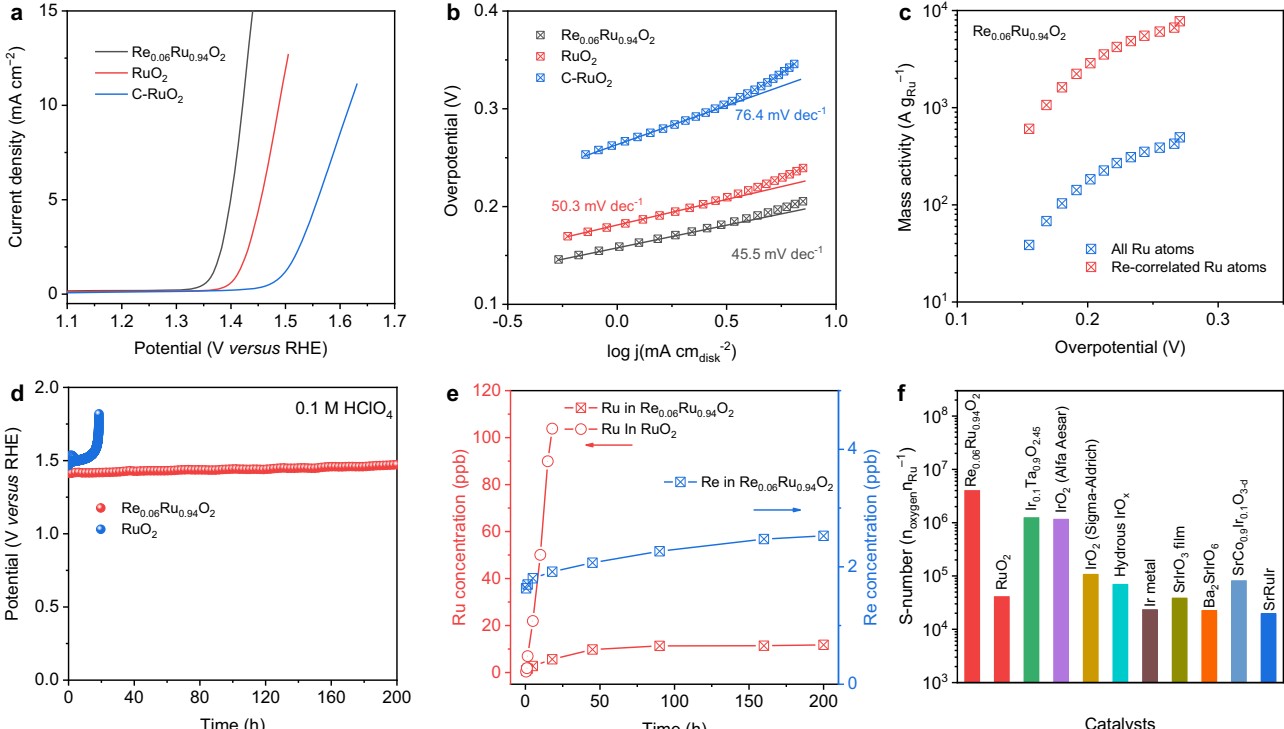

**Fig. 1 | OER performance. a** LSV curves and **b** Tafel plot for $Re_{0.06}Ru_{0.94}O_2$, $RuO_2$, and commercial $RuO_2$ in $O_2$-saturated 0.1 M $HClO_4$ (RHE = reversible hydrogen electrode). **c** Comparison of mass activity for Ru atoms in $Re_{0.06}Ru_{0.94}O_2$ as a function of overpotential. **d** Constant current chronopotentiometric stability measurements at anodic current density 10 mA cm$^{-2}$ for $Re_{0.06}Ru_{0.94}O_2$ and $RuO_2$. **e** Dissolved Ru (left ordinate - $y$ axis) and Re (right ordinate - $y$ axis) ion concentration in electrolyte for $Re_{0.06}Ru_{0.94}O_2$ and $RuO_2$ determined via ICP-MS. **f** S-number for $Re_{0.06}Ru_{0.94}O_2$ and $RuO_2$ catalyst and Ir-based OER catalysts in acid.

turnover frequency (TOF)[39] for $Re_{0.06}Ru_{0.94}O_2$ at overpotential 272 mV is 0.17 s$^{-1}$ (Supplementary Fig. 8a), which is an order of magnitude greater than for C-RuO$_2$ of 0.004 s$^{-1}$. Because recorded current might be affected by the Re and Ru reconstruction, we measured the Faradaic efficiency (FE) for OER by real-time monitoring O$_2$. The $Re_{0.06}Ru_{0.94}O_2$ exhibited a FE of ~100% at current density from 5 to 25 mA cm$^{-2}$, confirming high OER selectivity (Supplementary Fig. 8b). A comparative performance of $Re_{0.06}Ru_{0.94}O_2$ with reported OER electrocatalysts is presented in Supplementary Table 1. It is seen from the table that $Re_{0.06}Ru_{0.94}O_2$ exhibits comparatively excellent OER performance in acidic electrolytes, better than other recently reported noble metal-based electrocatalysts. The doping impact of Re on OER performance was determined (Supplementary Figs. 9–14) via a series of Re-RuO$_2$ with different Re. It was found that Re doping does not measurably change rutile structure for Re-RuO$_2$.

Durability of $Re_{0.06}Ru_{0.94}O_2$ catalyst is an important parameter for acidic OER electrocatalysts[40,41]. Figure 1d presents the chronopotentiometry data for $Re_{0.06}Ru_{0.94}O_2$ at constant current density 10 mA cm$^{-2}$ via loading on carbon paper with a loading mass 0.2 mg$_{cata}$ cm$^{-2}$. It is seen that the potential for $Re_{0.06}Ru_{0.94}O_2$ remained steady (constant) for a continuous 200 h test. However, RuO$_2$ exhibited a rapid activity decay within 19 h, likely the result of Ru dissolution in the acidic electrolyte. Importantly, carbon paper is not an ideal support for acidic OER durability testing because of due substrate passivation[42–46]. As a result, chronopotentiometry is not always a reliable technique to determine stability of acidic OER catalysts on carbon paper. Detecting catalyst mass losses during OER can provide quantitative information that distinguishes between different degradation mechanisms[42]. Therefore, Ru dissolution in different catalysts was determined to confirm a degradation mechanism. Figure 1e and Supplementary Table 2 present the time-dependent Ru and Re concentration in the electrolyte, corresponding to Fig. 1d. Dissolved Ru concentration for RuO$_2$ increased rapidly to 104 ppb within 20 h, equaling 2.7% Ru loss. In contrast, dissolved Ru and Re for $Re_{0.06}Ru_{0.94}O_2$ were significantly less at 11.8 and 2.5 ppb after stability testing for 200 h. By converting to mass loss of the metal specie, there was just 0.34% of loss with Ru, and 0.62% with Re. In addition, Ru and Re dissolution rates in $Re_{0.06}Ru_{0.94}O_2$ decreased over time, evidencing a stable structure during OER. The stability number (S-number) for the catalysts was determined via measuring oxygen produced and dissolved metal ion concentration in the electrolyte (Supplementary Fig. 15)[40]. Both RuO$_2$ and $Re_{0.06}Ru_{0.94}O_2$ show a time-dependent S-number, similar to the recently reported work[47]. The S-number of the catalysts increases with time, and is attributed to the slower Ru dissolution rate. Significantly, the S-number for $Re_{0.06}Ru_{0.94}O_2$ is significantly greater than that reported for Ru-based electrocatalysts and, importantly, comparable with Ir-based catalysts (Fig. 1f)[28,37,40].

### Structural evolution of $Re_{0.06}Ru_{0.94}O_2$

The stability of $Re_{0.06}Ru_{0.94}O_2$ was determined via crystal structure and chemical states of samples after 50 h chronopotentiometric test in 0.1 M HClO$_4$. Re single atoms are anchored in the RuO$_2$ crystal without aggregation or reconstruction after OER, as confirmed in the aberration-corrected high-angle annular dark-field scanning transmission electron microscopy (HAADF-TEM) images in Fig. 2a, b, evidencing the stability of Re dopants in the rutile structure during catalyzing. Corresponding elemental mapping confirmed that Re atoms are distributed uniformly in $Re_{0.06}Ru_{0.94}O_2$ nanoparticles, similar to that for the pristine samples (Fig. 2c). Significantly, RuO$_2$ also exhibited a rutile structure after 20 h OER testing (Supplementary Fig. 16), evidencing that degradation of active site does not change crystal structure of the matrix meaningfully. X-ray photoelectron spectroscopy (XPS) and synchrotron-based near-edge X-ray absorption fine structure (NEXAFS) spectroscopy were used to determine the electronic state for $Re_{0.06}Ru_{0.94}O_2$ and RuO$_2$ prior to and after OER testing. As is presented in Supplementary Figs. 5 and 17a, the valence state for Ru in RuO$_2$ after OER increased when compared with that for pristine RuO$_2$, whereas Ru sites in $Re_{0.06}Ru_{0.94}O_2$ were more stable than in RuO$_2$. In addition, the Re sites in $Re_{0.06}Ru_{0.94}O_2$ were highly stable during OER (Supplementary Fig. 17b). O-related spectra for RuO$_2$ evidence higher average Ru valence states and charge redistribution compared with pristine sample, that is caused by Ru dissolution-induced catalyst degradation (Supplementary Figs. 17c, d)[48,49].

Ru K-edge and Re L$_3$-edge XAS prior to and after 50 h OER were characterized to determine the local coordination environment-change of $Re_{0.06}Ru_{0.94}O_2$ after stability testing. Compared with pristine $Re_{0.06}Ru_{0.94}O_2$, the Ru K-edge X-ray absorption near-edge spectroscopy (XANES) of the sample after OER exhibited an (apparently) unchanged valence state with dominated Ru$^{4+}$ (Fig. 2d), consistent with XPS results. Besides, Fourier-transformed magnitudes of the Ru K-edge extended X-ray absorption fine structure (EXAFS), which exhibited two main peaks at ~1.5 and 3.1 Å, showed slight change in $Re_{0.06}Ru_{0.94}O_2$ (Fig. 2e). They corresponded to the nearest-neighbor Ru−O and next nearest-neighbor Ru−Ru/Re coordination shells, respectively, as confirmed by EXAFS wavelet transformed (WT) analysis which reveals two intensity maxima at ~4.2 and 8.0 Å$^{-1}$ (Fig. 2f). Further EXAFS fitting showed that the Ru−O peak can be divided into two distinct sub-shells with interatomic distances of 1.92 Å (coordination number $N$ is 1.8) for Ru−O1 and 2.01 Å (coordination number $N$ is 4.1) for Ru−O2, similar to the pristine sample (Supplementary Fig. 18 and Supplementary Table 3). In addition, the apparently unchanged coordination number excludes the formation of highly concentrated Ru/Re vacancies. Therefore the coordination environment for Ru in $Re_{0.06}Ru_{0.94}O_2$ following OER is confirmed to be unchanged, evidencing that Re doping strengthens the Ru−O bond and prevents Ru dissolution. In addition, Re L$_3$-edge XANES confirms that $Re_{0.06}Ru_{0.94}O_2$ after OER exhibits a similar Re valence state (6.33) to the pristine sample (6.38) (Fig. 2g and Supplementary Fig. 19). The Re L$_3$-edge FT-EXAFS spectrum and WT-EXAFS for $Re_{0.06}Ru_{0.94}O_2$ prior to and after OER demonstrate a rather similar profile to the Ru K-edge (Fig. 2h, i), confirming that the Re sites are stable in the RuO$_2$ matrix without formation of ReO$_x$-related species (Supplementary Fig. 20 and Supplementary Table 4). It should be noted that the FT-EXAFS of Ru K-edge and Re L$_3$-edge shows similar shape, confirming the substitutional doping of Re atoms in RuO$_2$.

### Dynamic electron transferring in $Re_{0.06}Ru_{0.94}O_2$

Because post-reaction characterizations cannot be used to determine the change in materials during OER, operando Re L$_3$-edge XAS at different overpotentials were carried out to understand the interaction between Re and sites in situ. Changes in the local electronic and atomic structures from open-circuit potential (OCP) to 1.6 V were determined via XANES and EXAFS analysis. As is shown in Fig. 3a–c, the Re dopants in $Re_{0.06}Ru_{0.94}O_2$ exhibit dynamic electron accepting-donating, which, importantly, is different from conventional dopants. Prior to OER, the valence state of Re increased because of Re oxidation with the increase of oxidizing potential. At overpotentials around OER on-site, the Re valence decreased significantly to less than the pristine state. However, at large overpotential, the valence state of Re again increased. This dynamic electron transfer only occurs with applied potential, and is not observed in the post-reaction XAS measurements, Fig. 2. The operando XAS measurement was repeated three times to confirm this unique dynamic electron transfer. Based on the spectral evolution for Re L$_3$-edge, the OER on $Re_{0.06}Ru_{0.94}O_2$ was divided into three stages, (1) pre-catalytic from OCP to 1.35 V, (2) on-site catalytic stage from 1.35 to 1.5 V and, (3) large-overpotential stage from 1.5 to 1.6 V. To determine reaction mechanism in situ the change of Re valence state (Fig. 3d) and the bond length for Re−O1 and Re−O2 (Fig. 3e, Supplementary Fig. 21, and Supplementary Table 4) were analyzed.

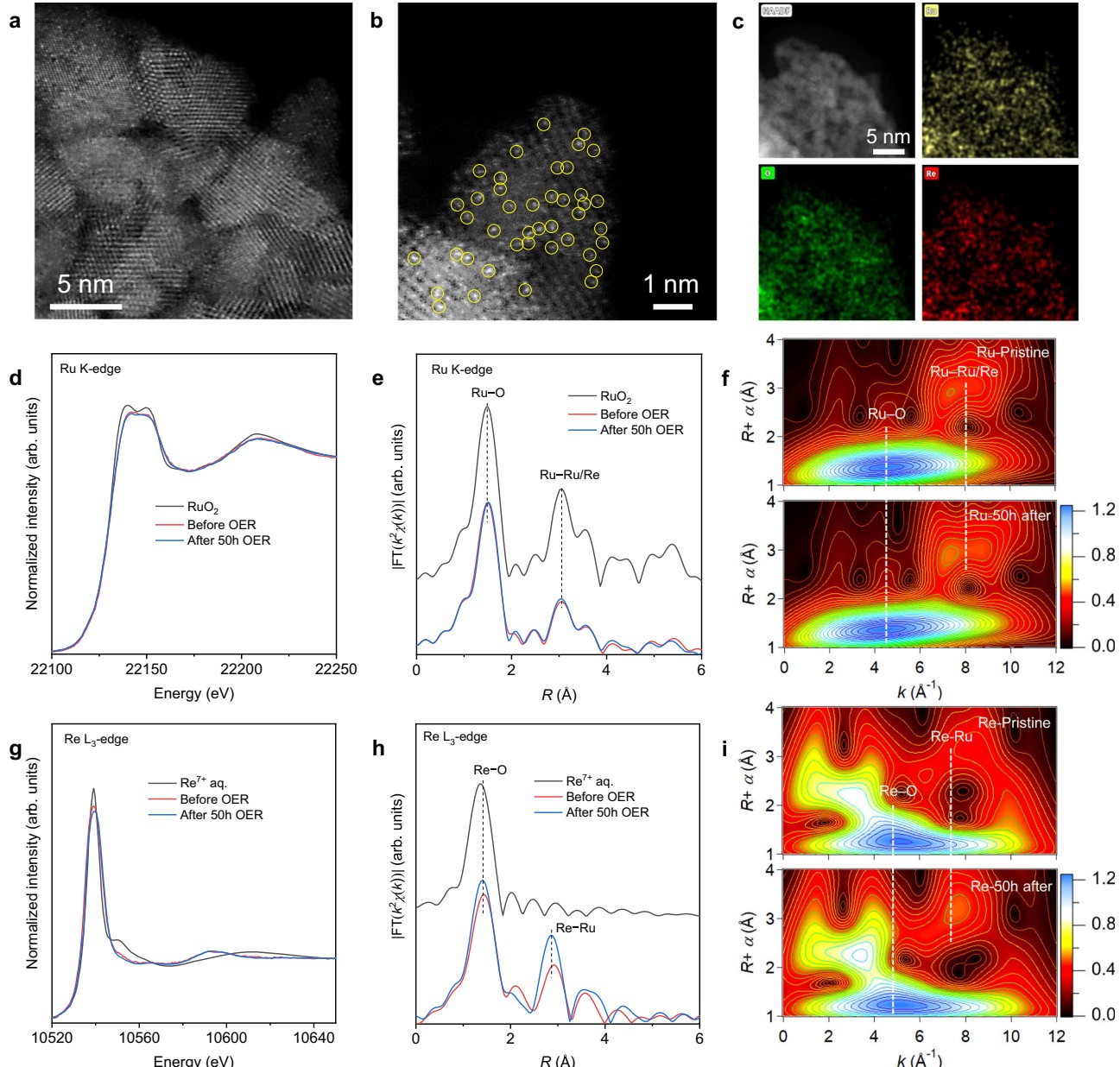

**Fig. 2 | Structural characterization of $Re_{0.06}Ru_{0.94}O_2$ after 50 h OER in acid.**
**a, b** Aberration-corrected HAADF-STEM image of $Re_{0.06}Ru_{0.94}O_2$ after 50 h OER. Re sites are labeled with yellow-color circles. **c** EDS mapping images of $Re_{0.06}Ru_{0.94}O_2$ catalyst after 50 h OER. **d, e** Ru K-edge XANES spectra and FT $k^2$-weighted EXAFS signals for $RuO_2$, $Re_{0.06}Ru_{0.94}O_2$ before and after 50 h OER. **f** Comparison of Ru K-edge WT-EXAFS for $RuO_2$, $Re_{0.06}Ru_{0.94}O_2$ before and after 50 h OER. **g, h** Re $L_3$-edge XANES spectra and FT $k^2$-weighted EXAFS signals for $Re^{7+}$ in aqueous solution, $Re_{0.06}Ru_{0.94}O_2$ before and after 50 h OER. **i** Comparison of Re $L_3$-edge WT-EXAFS for $Re_{0.06}Ru_{0.94}O_2$ before and after 50 h OER.

In Stage (1), the average valence state for Re of $Re_{0.06}Ru_{0.94}O_2$ increased from 6.33 to 6.67 when compared with the pristine sample (Fig. 3a, d), while a similar intensity increase was apparent in the first peaks in FT-EXAFS spectra (red-color region, Fig. 3b). In particular, with applied potential increased from OCP to 1.3 V, the Re–O peak shifted positively from 1.36 to 1.38 Å. To determine the reason for these changes, WT analysis of EXAFS spectra was conducted, which provides R- and k-space information and discriminates the back-scattering atoms (Fig. 3c). For the contour intensity maximum corresponding to the FT-EXAFS peak for Re–O at 1.36 Å, an evident intensity increase is observed at 9.0 Å$^{-1}$ (denoted by a dashed line in Fig. 3c), which agrees well with the location of Re–Ru scattering. These changes confirm that this stage is pre-catalytic, in which the increased valence state of Re is due to the oxidizing potential, and not OER reaction.

With an applied potential >1.3 V, Stage (2), the most apparent feature is that the Re valence state decreased significantly from 6.67 to 6.29, as is shown in Fig. 3d. In addition, the bond length for Re–O2 elongates obviously with the increase of applied potentials, evidencing a dynamic electron-transfer amongst Ru, Re and adsorbed oxygen intermediates (Fig. 3e), demonstrating that Re dopants gain electrons from Ru site to tune electronic structure, and activate Ru sites. It is widely acknowledged that the high-valence Ru species are the active sites for OER. Therefore, the Re gains electrons from Ru at Stage (2), to facilitate the formation of high-valence Ru sites to boost OER. This stage also explains why the activity of $Re_{0.06}Ru_{0.94}O_2$ is greater than pure $RuO_2$.

In Stage (3), with the applied potential reaching 1.5 V, the Re valence state increases again from 6.29 to 6.53 (Fig. 3d), evidencing

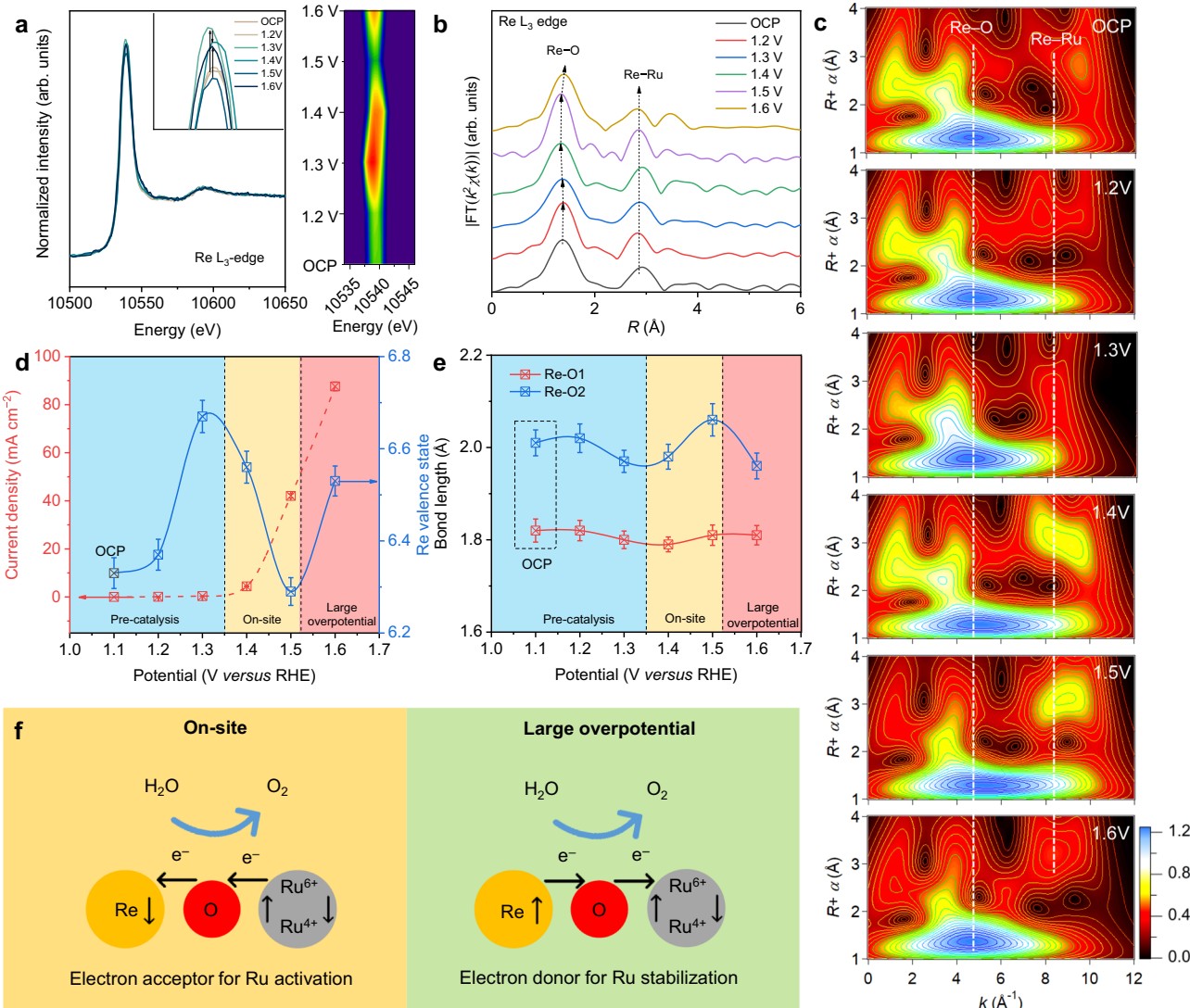

**Fig. 3 | Operando XAS characterization of Re$_{0.06}$Ru$_{0.94}$O$_2$ confirming dynamic process. a** Re L$_3$-edge XANES spectra for Re$_{0.06}$Ru$_{0.94}$O$_2$ at differing potentials in O$_2$-saturated 0.1 M HClO$_4$. Right part: contour plot of the Re L$_3$-edge white peak intensity. **b** FT-EXAFS signals for Re$_{0.06}$Ru$_{0.94}$O$_2$ corresponding to (**a**). **c** Comparison of Re L$_3$-edge WT-EXAFS plots for Re$_{0.06}$Ru$_{0.94}$O$_2$ from OCP to 1.6 V. **d** Change in Re valence state and OER current as a function of applied potential. **e** Change in bond length for Re−O1 and Re−O2 coordination shells. The operando XAS measurements were repeated three times to produce the error bar. **f** Schematic for dynamic electron transfer in Re$_{0.06}$Ru$_{0.94}$O$_2$. In OER on-site region, Re dopants gain electrons from neighboring Ru to boost OER. At large overpotential, Re dopants donate electrons back to Ru and prevent dissolution of active sites.

that the Re dopants donate the electrons back to Ru active site. Importantly, WT-EXAFS analysis highlights the appearance of the Ru−Re scattering signal at 1.3 V, and its disappearance at 1.6 V, to demonstrate a dynamic bonding length change between Re−Ru coordination shell, which is associated with a contraction of Re−O2 bond length. This stage evidences that the Re dopants protect the active site from dissolution at large overpotential by donating the electrons back to Ru to maintain stable catalyzing.

We investigated the Ru sites via operando Ru K-edge XAS to determine the change for Ru cations. In contrast to dynamic change for Re cations with applied potential, it was found that both operando Ru K-edge XANES and EXAFS exhibit only 'slight' change during catalysis (Supplementary Fig. 22), as was confirmed by detailed Ru K-edge EXAFS fitted analyses (Supplementary Figs. 23−24 and Supplementary Table 3). It is widely acknowledged that XAS analyses reflect average information for all Ru atoms in Re$_{0.06}$Ru$_{0.94}$O$_2$. The XAS signal for Ru−O−Ru site dominates and overlaps that for Ru−O−Re site because of the low doping amount of Re in Re$_{0.06}$Ru$_{0.94}$O$_2$, leading to the unchanged

Ru K-edge spectra. Therefore, we focused mainly on the Re L$_3$-edge to determine the dynamic behavior of Ru−O−Re sites.

Operando XAS results confirm that the electron transfer between Re and Ru site is potential-depended in which the function for Re varies. As is summarized in Fig. 3f, the Re$_{0.06}$Ru$_{0.94}$O$_2$ electrocatalyst undergoes a three-step dynamic electron transfer during OER, which is different from conventional RuO$_2$ (Supplementary Fig. 25). At the pre-catalysis stage, the increase in Re valence state is due to the gradually increased oxidizing potential. During OER, Stage (2), the Re atoms gain electrons from Ru site via the O bridge to boost Ru sites for catalyzing, which explains the boosted activity of Re$_{0.06}$Ru$_{0.94}$O$_2$. At large overpotentials, Stage (3), the Re dopants donate electrons back to prevent over-oxidation of Ru active sites and formation of H$_2$RuO$_5$ species. Therefore, the stability of Re$_{0.06}$Ru$_{0.94}$O$_2$ is significantly boosted to greater than most Ir-based electrocatalysts. Consequently, it is concluded that the Re dopants act as a dynamic electron reservoir that achieves an adaptive tune of the OER active site in situ.

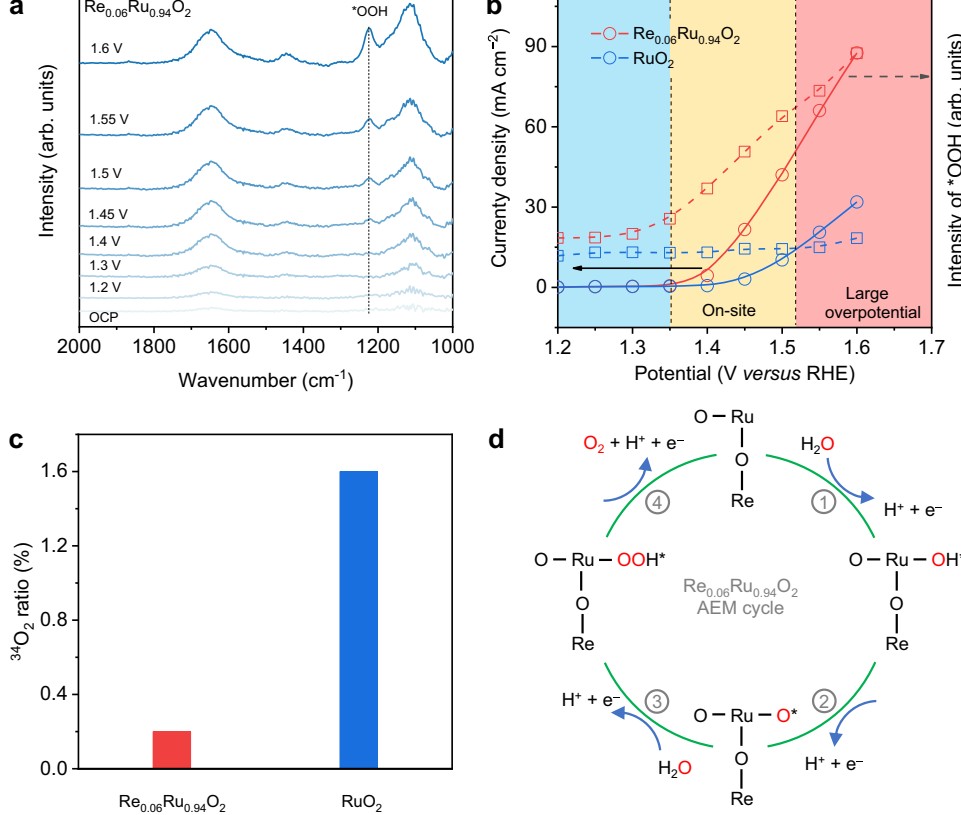

**Fig. 4 | Reaction mechanism on $Re_{0.06}Ru_{0.94}O_2$ and $RuO_2$. a** In situ ATR-SEIRAS spectra for $Re_{0.06}Ru_{0.94}O_2$ during multi-potential steps. **b** Potential dependence of band intensity of characteristic vibration adsorption of surface-adsorbed *OOH. **c** $^{34}O_2$ ratio determined via normalization of DEMS signal for $Re_{0.06}Ru_{0.94}O_2$ and $RuO_2$ in 0.05 M $H_2SO_4$-$H_2^{16}O$. **d** Illustration for AEM pathway on $Re_{0.06}Ru_{0.94}O_2$ toward acidic OER.

## Mechanistic analysis of dynamic electron transfer

The boosted stability of most acidic OER electrocatalysts is from the change in reaction pathway from LOM to AEM[28,31,50]. This is because the AEM pathway involves multiple intermediates of *OH, *O, and *OOH without lattice oxygen participation, leading to long-term catalyzing[51]. Therefore, additional in situ measurements were conducted to determine the impact of dynamic electron transfer on the reaction mechanism for $Re_{0.06}Ru_{0.94}O_2$. It is widely acknowledged that conventional $RuO_2$ catalyzes OER in a LOM-involved pathway (Supplementary Fig. 26)[16,52] in which the mobilized crystal structure is caused by the oxygen vacancies and leached Ru site. However, the $Re_{0.06}Ru_{0.94}O_2$ exhibited significantly less Ru dissolution than $RuO_2$, most probably because of the changed reaction pathway from LOM to AEM.

We used in situ attenuated total reflectance surface-enhanced infrared absorption spectroscopy (ATR-SEIRAS) to confirm the OER mechanism on $Re_{0.06}Ru_{0.94}O_2$. This is because this technique can examine the potential-dependent surface reaction intermediates. The in situ ATR-SEIRAS spectra for $Re_{0.06}Ru_{0.94}O_2$ at differing working potential (Fig. 4a) exhibit a distinct absorption peak at 1224 $cm^{-1}$, which is attributed to the O−O stretching of surface-adsorbed *OOH, a typical intermediate for AEM pathway[53]. Peak intensity increased linearly from OER on-site to 1.6 V, evidencing a constant AEM pathway at different catalyzing stages (Fig. 4b). In comparison, $RuO_2$ exhibits unidentifiable *OOH with weaker intensity after OER on-site (Fig. 4b and Supplementary Fig. 27), confirming a LOM-involved pathway. Noted that an identifiable *OOH peak appeared at an applied potential on $RuO_2$ of 1.6 V (Supplementary Fig. 28), evidencing that a combined LOM-AEM pathway dominates the reaction at large overpotential. In addition, the LSV curves for $Re_{0.06}Ru_{0.94}O_2$ and $RuO_2$ (Supplementary Fig. 29) were carefully analyzed. The $RuO_2$ exhibited an apparent activation-

deactivation that agreed well with features of the LOM pathway. In contrast, $Re_{0.06}Ru_{0.94}O_2$ exhibited stable activity under cycling, evidencing a stable AEM characteristic.

To further confirm the AEM reaction pathway on $Re_{0.06}Ru_{0.94}O_2$ and $RuO_2$, $^{18}O$ isotope-assisted operando differential electrochemical mass spectrometry (DEMS) analyzes were carried out, which can directly differentiate AEM and LOM. Samples were first labeled using $H_2^{18}O$- contained electrolytes (Supplementary Figs. 30–31). In a typical LOM pathway, the lattice oxygen labeled with $^{18}O_{ads}$ couples with $^{16}O$ in the electrolyte to generate $^{34}O_2$. In contrast, the AEM pathway produces $^{36}O_2$ from water-splitting because no lattice oxygen participates in OER[28]. As is shown in Fig. 4c and Supplementary Fig. 32, $RuO_2$ produces a clear $^{34}O_2$ signal with high intensity, significantly greater than that for $Re_{0.06}Ru_{0.94}O_2$. Given that both catalysts exhibit a similar ECSA (Supplementary Fig. 10), the influence of physically adsorbed $H_2^{18}O$ is excluded. Quantitatively, the $RuO_2$ produces 1.6% $^{34}O_2$, evidencing a LOM-contained pathway. However, the $Re_{0.06}Ru_{0.94}O_2$ produces just 0.3% $^{34}O_2$, close to the $^{18}O$ content in natural water and air, evidencing the AEM pathway (Fig. 4d). Importantly, it was reported that <0.2% of evolved oxygen contains an oxygen atom originating from $RuO_x$[11]. The difference with our findings is because of poor crystallinity of our sample obtained from the molten salt method, which has more active lattice oxygen. It is concluded that the in situ FTIR, online DEMS and post-reaction XPS measurements confirm the reaction pathway on $Re_{0.06}Ru_{0.94}O_2$ and $RuO_2$. The dynamic Re dopants change the reaction pathway on $Re_{0.06}Ru_{0.94}O_2$ from LOM to AEM, leading to boosted OER.

## Computations for activity and stability origin

DFT computations were performed to provide qualitative analyses of the OER mechanism and the stability origin of $Re_{0.06}Ru_{0.94}O_2$. Based

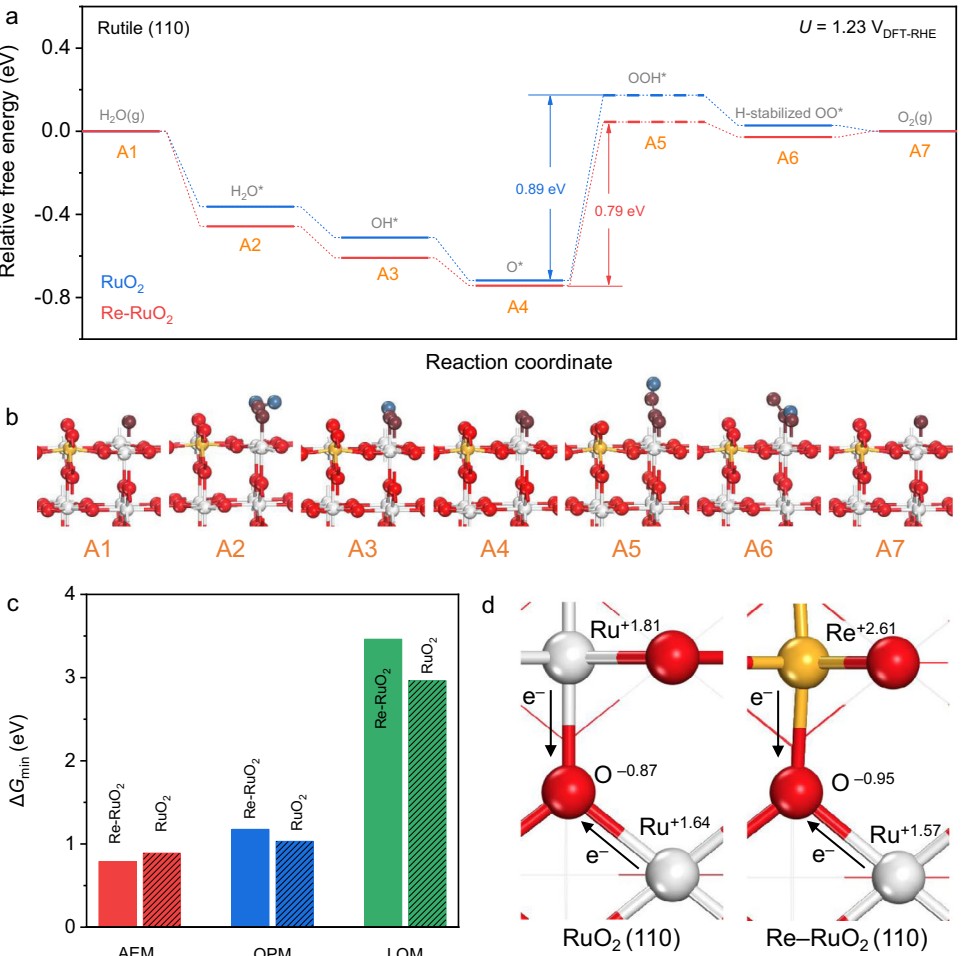

**Fig. 5 | DFT simulation. a, b** Free energy diagram for OER on unsaturated Ru site of $Re_{0.06}Ru_{0.94}O_2$ and $RuO_2$ at 1.23 V *versus* RHE, showing six (6) possible intermediates for the (110) surfaces. Dashed lines indicate unstable −OOH precursor states, shown as H-stabilized OO*. **c** Minimum activation energy for different reaction pathways for $Re_{0.06}Ru_{0.94}O_2$ and $RuO_2$. **d** Oxidation state for $RuO_2$ and $Re_{0.06}Ru_{0.94}O_2$ described by differences in electron transfer based on Bader charge computations. Black-color arrows show direction of electron transfer.

on the STEM and X-ray powder diffraction (XRD) findings, a Re-doped rutile $RuO_2$ (110) slab model was constructed. Various doping sites were tested, and the most stable structure was selected. Figure 5a presents the OER with AEM pathway on $RuO_2$ and Re-$RuO_2$ (110), based on in situ measurements. The unsaturated Re site on Re-$RuO_2$ was inactive for OER because of the too strong adsorption of key oxygen intermediates (Supplementary Fig. 33). Reaction intermediate configurations on Re-$RuO_2$ (110) are illustrated in Fig. 5b (Those for $RuO_2$ are shown in Supplementary Fig. 34). The initial step is the formation of a deoxygenated surface (A1), followed by adsorption of a water molecule on the Ru site (A2) together with subsequent formation of OH* (A3) and O* (A4) species from $H_2O*$ deprotonation. Then, the adsorption of another water molecule occurs to compose an OOH* (A5). It should be noted that this step is unstable, in which the OOH* donates the proton to the neighboring oxygen to form an H-stabilized OO* species (A6). Molecular oxygen is then formed from this H-stabilized OO* (A7)[52]. As is shown in Fig. 5a, the step from A4 to A5 is the most energetically difficult step under 1.23 $V_{DFT-RHE}$ (defined by a computational hydrogen electrode model)[54], where Re-$RuO_2$ exhibits reaction energy of 0.79 eV, 0.1 eV less than that on $RuO_2$. Thermodynamic advantage of Re-doped $RuO_2$ over pure $RuO_2$ is therefore demonstrated. In addition to LOM and AEM pathways, a new oxide path mechanism (OPM) pathway has been confirmed that allows direct O−O radical coupling without generation of oxygen vacancy defects and extra reaction intermediates (Supplementary Fig. 35)[55,56].

Therefore, Fig. 5c compares the minimum required energy to activate AEM, LOM, and OPM pathways on, respectively, $Re_{0.06}Ru_{0.94}O_2$ and $RuO_2$. The AEM pathway is seen in the figure to be the most energetically favorable compared with the other two on both surfaces, evidencing that it dominates OER under the equilibrium potential. Qualitative analyses of OER mechanism for $RuO_2$ and Re-$RuO_2$ with metal vacancies were assessed via DFT computation. Importantly, the structure for Re-$RuO_2$ with a Re vacancy is the same as for $RuO_2$ with a Ru vacancy following stabilization. We considered Re (vac-$RuO_2$) and Ru (vac-Re-$RuO_2$) vacancies therefore on Re-$RuO_2$ to determine the impact on electrocatalytic performance. As is seen in Supplementary Figs. 36 and 37, although the sample with metal defects exhibits optimized OPM thermodynamic energy, it is less than that for AEM on Re-$RuO_2$, confirming the advantage of Re-$RuO_2$ over $RuO_2$ or metal defects.

Although the in situ measurements appear to evidence that OER on $RuO_2$ is likely to occur through a combined LOM-AEM pathway, we hypothesize that the LOM pathway on $RuO_2$ is activated because of the applied potential and corrosive acidic environment that mobilizes the Ru−O bond and leads to the generation of O vacancies. This greater tendency for the AEM pathway with Re-$RuO_2$ and stability origin was examined via surface oxidation states. As is seen from Bader charge analysis (Fig. 5d), more negative charge is transferred from Re to Ru via the O bridge on the Re-$RuO_2$ surface compared with $RuO_2$, confirming that Re doping reinforces the stability of Ru−O bond, and promotes

the AEM under acidic media. An acknowledged drawback, however, is that DFT computations cannot simulate the dynamic electron transfer because the bond length and electronic structure differ with applied potential change. However, the DFT does provide a reasoned, qualitative evaluation of $Re_{0.06}Ru_{0.94}O_2$ that is important to establish the activity and stability origin of the catalysts.

## Discussion

In summary, we report a $Re_{0.06}Ru_{0.94}O_2$ catalyst with dynamic Re dopants that exhibits significantly boosted activity and stability for acidic OER. Interaction between aciduric Re dopant and Ru active site is tuned by dynamic electron-accepting-donating in which the Re gains electrons to activate the Ru site at on-site overpotential and donates electrons back at large overpotential to prevent Re dissolution. Consequently, $Re_{0.06}Ru_{0.94}O_2$ exhibits intrinsic activity and corrosion resistance to make amongst the best acidic OER electrocatalysts. Based on combined, judicious in situ measurement and DFT computation, we confirm that boosted catalytic activity and corrosion resistance originates from modified electronic structure via dynamic electron transfer, facilitating the change of pathway on $Re_{0.06}Ru_{0.94}O_2$ from LOM to AEM and lessens adsorption energies for oxygen intermediates in the active site. We conclude that dynamic dopants can be used to boost the activity and stability of active sites and therefore guide the design of adaptive electrocatalysts for improved clean energy conversions.

## Methods

### Chemicals

Sodium nitrate (≥99%), ruthenium (III) chloride hydrate (ruthenium content, 40.00 to 49.00%), and ruthenium (IV) oxide (≥99.9%) were purchased from Sigma-Aldrich without further purification. Sodium perrhenate (99%) was purchased from Alfa Aesar, Australia. $^{18}O$ Water ($^{18}O$, 97%+) was purchased from Novachem, Australia.

### Catalyst synthesis

$Re_{0.06}Ru_{0.94}O_2$ were synthesized via a molten-salt method. Typically, 5 g $NaNO_3$ and 100 mg $NaReO_4$ were mixed in a 50 mL porcelain crucible. The crucible was transferred to a muffle furnace and heated to 360 °C for 10 min. 40 mg $RuCl_3·xH_2O$ power was added to the crucible (with lid) and held for 5 min. $NO_2$ brown-color gas was generated when adding $RuCl_3·xH_2O$ because of oxidation of $Ru^{3+}$. The crucible was removed from the furnace and cooled to room temperature (25 °C) under ambient conditions. The product was washed with deionized water to remove salt and obtained via vacuum filtration (Millipore 0.22 μm). $RuO_2$ was synthesized without addition of $NaReO_4$. $Re_{0.03}Ru_{0.94}O_2$, $Re_{0.04}Ru_{0.94}O_2$, $Re_{0.05}Ru_{0.94}O_2$, $Re_{0.06}Ru_{0.94}O_2$, and $Re_{0.1}Ru_{0.9}O_2$ were prepared via varying $NaReO_4$ to, respectively, 20, 40, 80, and 150 mg.

### Electrochemical measurement

Typically, 2 mg of catalyst was dispersed in 950 μL of deionized water. 50 μL 5 wt% of Nafion/water was added to the catalyst dispersion. 10 μL of the catalyst dispersion (2 mg mL$^{-1}$) was dropped onto a 5 mm glassy-carbon rotating disk electrode (0.1 mg$_{cata}$ cm$^{-2}$) serving as the working electrode. The reference electrode was Ag/AgCl in 3.5 M AgCl–KCl solution, and the counter electrode was a graphite-rod. The reference electrode was calibrated in hydrogen-saturated 0.01 M $HClO_4$. The calibrated value for Ag/AgCl in 3.5 M AgCl–KCl was equal to 0.214 (close to the theoretical value of 0.205). To evaluate OER for different catalysts, the working electrode was put through 10 cyclic voltammetry (CV) scans between 1.1 to 1.6 V at a 20 mV s$^{-1}$ to clean and stabilize the surface of the catalyst. All potentials were referenced to RHE via adding (0.214 + 0.059 × pH) V, and all polarization curves were corrected for iR compensation within the cell. A flow of $O_2$ was maintained over the electrolyte during experiment. The working electrode was rotated at 1600 rpm to remove $O_2$ gas formed on the catalyst

surface. Electrochemical data were corrected for uncompensated series resistance $R_s$ that was determined via open-circuit voltage. A 90% i-R compensation was selected to determine $R_s$. The final polarization curve was determined via i-R compensation. To determine metal dissolution in the electrolyte, a stability test was conducted in an H-cell. The stability measurements were carried out by air-brush spraying the catalysts uniformly on the carbon with a loading mass of 0.2 mg$_{cata}$ cm$^{-2}$. The stability test was conducted under 25 °C. The cathode and the anode chamber were separated by a Nafion film to obviate re-deposition of Ru/Re on the counter electrode. Carbon paper (effective area 1 cm$^{-2}$) was used as working electrode with a loading mass of 0.2 mg$_{cata}$ cm$^{-2}$.

### In situ ATR-SEIRAS measurement

In situ ATR-SEIRAS was carried out with a Nicolet iS20 spectrometer with an HgCdTe (MCT) detector cooled with liquid $N_2$ and a VeeMax III (PIKE Technologies) accessory. Electrochemical tests were conducted in a custom-made, three-electrode electrochemical single-cell. A Pt-wire and a saturated Ag/AgCl were used as, counter and reference electrodes. A fixed-angle, Au-coated Si prism (60°) was used to load catalysts and served as the working electrode. Au thin-layer with a thickness 40 nm was coated by magnetron sputtering. The reflecting plane of the Si prism was polished with diamond compound (0.05 μm, Kemet. Int. Ltd.), and sonicated in acetone, ethanol, and water before sputtering. For ATR-SEIRAS measurement, 32 scans were collected with a spectral resolution of 4 cm$^{-1}$ for each spectrum. The background spectrum of the working electrode was recorded in an open-circuit condition. The recorded spectra were processed via OMNIC software.

### Online DEMS with isotope labeling

The online DEMS system used was HPR-40, HIDEN. Measurements were determined using a modified DEMS cell (cell A) at atmospheric pressure. The pressure difference between the cell and the vacuum chamber allows oxygen species to be drawn into the vacuum chamber for mass spectrometer analysis. The working electrode is a Au-film with 50 nm thickness that is sputtered on a hydrophobic polytetrafluoroethylene membrane with a pore size 0.02 μm (Hangzhou Cobetter Filtration Equipment Co.). Catalyst ink was dropped into the Au-film, and dried. For isotope labeling, 3 mL of 0.05 M $H_2SO_4$ was prepared using $H_2^{18}O$ (97%+) as solvent. Concentrated $H_2SO_4$ of 98%+ was used and not 70% $HClO_4$ to reduce any contamination of $H_2^{16}O$ in the solute. $Re_{0.06}Ru_{0.94}O_2$ and the $RuO_2$ were subjected to four (4) CV cycles in the potential range, respectively, 1.06 to 1.38 V and 1.1 to 1.42 V versus RHE at a scan rate 5 mV s$^{-1}$ for labeling catalyst surface with $^{18}O$, while mass signals for gaseous products $^{32}O_2$, $^{34}O_2$, and $^{36}O_2$ were recorded. The catalysts were washed with abundant water ($H_2^{16}O$) and dried in a vacuum-oven for 1 h to remove $H_2^{18}O$ molecules physically attached to the catalyst-layer, the $^{18}O$-containing species chemically bonded on the surface, remained. Catalysts with isotope-labeled surfaces were operated in a standard electrolyte 0.5 M $H_2SO_4$ with $H_2^{16}O$ as solvent. The mass spectrometer was used to monitor gaseous products, including $^{32}O_2$, $^{34}O_2$, and $^{36}O_2$. Prior to electrochemical measurement, all the electrolytes were purged with high-purity Ar to remove dissolved oxygen.

### Ex situ and operando XAS characterizations

Re $L_3$-edge and Ru K-edge XAS data were collected at beamlines 1W1B in the Beijing Synchrotron Radiation Facility (BSRF, operated at 2.5 GeV with a maximum current 250 mA) with a Si (111) double-crystal monochromator[57]. Ex situ XAS measurements were conducted in transmission mode at 25 °C. We used standard $N_2$-filled ion chambers to monitor the intensity of the incident and transmitted X-rays. Operando XAS measurements were carried out in the fluorescence model using a homemade in situ cell and an electrochemical workstation. The working electrode is $Re_{0.06}Ru_{0.94}O_2$ loaded on carbon

paper. In addition, Pt-wire was used as the counter electrode and Ag/AgCl electrode as the reference electrode. $O_2$-saturated 0.1 M $HClO_4$ solution was used as the electrolyte. The operando XAS measurements were repeated three times to confirm the trend and build an error bar. We used ATHENA and ARTEMIS modules implemented in IFEFFIT software to analyze XAS data[58]. We conducted background subtraction and normalization to obtain EXAFS data and applied a Hanning window to obtain $\chi(\mathbf{k})$ data Fourier-transformed to real ($R$) space. Least-squares curve-fitting of EXAFS $\chi(\mathbf{k})$ data was carried out in R-space to determine quantitative structural parameters around central atoms. The structural model of XAS was built based on the crystal structures for $RuO_2$ and Re-$RuO_2$, with scattering amplitudes, phase shifts, and photoelectron mean free path for all paths computed with ab initio code FEFF 8.6.

## Material characterization

X-ray powder diffraction (XRD) data were collected on a Rigaku MiniFlex 600 X-Ray diffractometer. Field-emission SEM imaging was determined on a FEI QUANTA 450 electron microscope. High-angle annular dark-field imaging, and EDS mapping, were determined on a FEI Titan Themis 80-200, operating at 200 kV.

The atomic ratio for Re over Ru was analyzed via ICP-MS, Agilent 7500cx instrument. The catalyst was dispersed and dissolved in aqua regia at temperature 25 °C prior to ICP-MS analysis. Dissolved Ru and Re in the electrolyte were analyzed via ICP-MS by diluting the electrolyte two times prior to measurement.

Synchrotron-based NEXAFS measurements were determined on the soft X-ray spectroscopy beamline at the Australian Synchrotron, which is equipped with a hemispherical electron analyzer and a microchannel plate detector to permit concurrent recording of the total electron, and partial electron, yield. Calibration of XPS data were normalized to the photoelectron current of the photon beam, measured on an Au-grid. Raw XANES data were normalized using Igor Pro 8 software.

## Computational details

All the structures were optimized by density functional theory (DFT) with a periodic plane-wave implementation using Vienna ab initio Package (VASP) code[57,59,60]. Perdew–Burke–Ernzerhof (PBE) functional within the generalized gradient approximation (GGA) was used to model the exchange-correlation energy[61]. The ionic cores were described by projector-augmented wave (PAW) pseudopotentials[62] with an energy cutoff of 500 eV. A first-order, Methfessel-Paxton smearing of 0.05 eV was applied to the orbital occupation during geometry optimization and for energy computations.

Next, the adsorption energies on different structures were evaluated using four-layer 2 × 2 supercells with the bottom two layers constrained. In addition, [6 × 6 × 1] Monkhorst-Pack $k$-point grids were also used with a convergence threshold of $10^{-5}$ eV for the iteration in the self-consistent field (SCF)[63]. Structural optimizations were kept until force components were <0.02 eV Å$^{-1}$. Phonon modules in VASP 5.3 code were used to compute the vibrational frequencies of free molecules and adsorbates. Free energy corrections, including correction of the effect from zero-point energy, pressure, inner energy, and entropy, were determined by applying a standard thermodynamic correction.

## Data availability

Data that support findings from this study are available from the corresponding author on reasonable request.

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

## Acknowledgements

This work was financially supported by the Australian Research Council (DP220102595 and FL170100154 received by S.-Z.Q.) and the Korea Institute of Energy Technology Evaluation and Planning (KETEP), the Ministry of Trade Industry & Energy (MOTIE) of the Republic of Korea [20203030040030]. H.J. gratefully acknowledges financial support from Institute for Sustainability, Energy and Resources, The University of Adelaide, Future Making Fellowship. J.D. acknowledges financial

support from the Youth Innovation Promotion Association of the Chinese Academy of Sciences. H.J. acknowledges Dr. Haijing Li for helpful discussion on XAS. NEXFAS measurements were undertaken on the soft X-ray beamline at the Australian Synchrotron. X.L. acknowledges financial support from National Natural Science Foundation of China (22109082). XAS spectra were determined at the 1W1B station in Beijing Synchrotron Radiation Facility (BSRF). SEM, ICP, and TEM measurements were conducted at Adelaide Microscopy, The Centre for Advanced Microscopy and Microanalysis.

## Author contributions

H.J. conceived the project. S.-Z.Q. supervised the project and whole studies. H.J., C.T., and H.Y. conducted synthesis and electrochemical measurements. H.J. conducted online DEMS and in situ ATR-FTIR measurements. Q.Z. and L.G. performed the TEM characterizations. P.A. and J.D. performed the ex situ and operando XAS measurements and analyzed data. X.L. and H.P. performed the DFT computations. Y.Z., T.S., and U.P. assisted analyzes of findings. H.J., H.Y., X.L., K.D., J.D., and S.-Z.Q. wrote the manuscript. S.-Z.Q. reviewed and edited the manuscript. All authors have approved the final version of the manuscript.

## Competing interests

The authors declare no competing interests.
