## [Peer review file · Nature Communications]

REVIEWER COMMENTS

Reviewer #1 (Remarks to the Author):

This manuscript reports Re doped RuO₂ (Re_{0.06}Ru_{0.94}O₂) catalyst that exhibits significantly improved activity and stability for acidic OER. The authors propose that during operation, the Re dopants in RuO₂ exhibits dynamic electron-accepting-donating behavior in which the Re gains electrons to activate the Ru site for a high OER activity and donates electrons back at large overpotential to maintain a high stability. The dynamic behaviour was followed mainly using operando Re L-edge X-ray absorption spectroscopy (XAS). Several other nice in-situ techniques such as Infrared absorption spectroscopy, differential electrochemical mass spectrometry (DEMS) were used as well, suggesting the OER mechanism for Re_{0.06}Ru_{0.94}O₂ is based on AEM, and the AEM renders the high stability of Re-RuO₂ catalyst in acidic conditions. Overall, the obtained results are interesting and this manuscript will be of broad interest and importance in Nat. Comm. However, there are a few issues to be clarified.

1. What is the spin configuration of Ru⁴⁺ in RuO₂? It is hard to understand the peak assignments for t_{2g} and e_g at O K edge XAS spectra shown in Supplementary Fig. 6 and Fig. 12
2. It is mentioned in page 7 that “The O related spectra for RuO₂ evidence an electron transfer from e_g to t_{2g} caused by Ru dissolution induced lattice collapse and catalyst degradation”. It should be noted that the peaks at O K-edge spectra represent the unoccupied state of transition metal cation. How can one evidence from the O XPS and K-edge XAS spectra that there is electron transfer from e_g to t_{2g}?
3. In page 7, “Further EXAFS fitting showed that the Ru-O peak ...(the coordination number N is 1.8)...”. Such a low coordination number of 1.8 seems not possible.
4. The EXAFS data fittings at Ru K edges (Supplementary Fig. 18), Re L edge (Supplementary Fig. 19 and Supplementary Fig. 20) do not look fitted well. More careful analysis and re-fitting the EXAFS data should be carried out. The description of the fitting procedure and the approximations used should be given in the Supplemental Materials.
5. The authors only show the dynamic changes of oxidization state and bonding length for Re cations using operando XAS. How do the Ru cations change accompanying the dynamic behavior of Re?
6. It is hard to see the difference in the Re L-edge from Figure 3a.

Reviewer #2 (Remarks to the Author):

In this work, the authors reported the Re dopants in RuO₂ that can undergo a dynamic electron accepting-donating, in this manner, the static electron redistribution induced by the conventional dopants can be well mitigated. Accordingly, the Re_{0.06}Ru_{0.94}O₂ was reported to show highly desired electrocatalytic performance with high mass activity and high stability. It is a piece of interesting work to develop viable electrocatalysts toward acidic OER, thus it can be accepted for publication in Nat Commun after suitable revision:

1 The sample Re_{0.06}Ru_{0.94}O₂ has a relatively low Re doping, how did the authors determine the atomic ratio between Re/Ru, in addition, is it possible to tune the ratio by adjusting the synthetic condition, and one may wonder if the synthesis of such Re_{0.06}Ru_{0.94}O₂ (with fixed ratio) is repeatable, moreover, it is interesting to study if the ratio will affect the electrocatalytic properties. Given the ratio of Re/Ru indeed varied during testing, will the variation of atomic ratio of Re/Ru degrade or promote the electrocatalytic performance?

2 The author will be appreciated if they can discuss the definition or difference regarding static electron redistribution and dynamic electron accepting-donating in the introduction section;

3 The LSV in Fig. 1a is obviously obtained by i-R compensations, the authors should detail the parameter of i-R compensations, and the LSV curves without i-R compensations are suggested to be supplied at least in the supporting information section so that the readers can see the practical status;

4 The doped Re atoms in Re_{0.06}Ru_{0.94}O₂ are highlighted by yellow circle in its corresponding Aberration corrected HAADF-STEM image (Fig. 2b), however, it is confusing why the authors can verify these dots are Re rather than Ru;

5 In this work, the activity and stability were evaluated by a different way from the previous paper, i.e., the mass activity and stability number of S-number, the area activity should be supplied as this parameter is more meaningful for performance evaluation, and please detail the description on how to calculate stability number of S-number.

Reviewer #3 (Remarks to the Author):

The manuscript by Jin et al. devotes to the study of ruthenium oxide dynamic doping by incorporating rhenium into the catalyst matrix. The work presents a large set of characterization techniques to support the claims that Re plays the role of a dynamic electron donor-acceptor, additionally promoting the adsorbate evolution mechanism. I believe that the work merits to be published in Nature Communications, although some aspects of the manuscript need to be improved.

Introduction: The authors aimed to summarize very important works in a very short section, and by doing so they fail to cite key literature in the field. See examples as follows:

- 1) Reference 8 is not the first report of RuO₄ detection: should be substituted by J. Electroanal. Chem. 1987, 237, 251-260
- 2) The RuO₂ hybridization with IrO₂ has been discussed in more relevant works: J. Electrochem. Soc. 2016, 163, F3099; ACS Appl. Energy Mater. 2020, 3, 5229-5237; ACS Catal. 2021, 11, 15, 9300-9316; Electrochim. Acta 2012, 70, 158-164; Appl. Catal. B: Environ. 2015, 164, 488-495.
- 3) Besides that, works by Chorkendorff and co-workers (Chem. Sci., 2015, 6, 190; Energy Environ. Sci., 2022,15, 1988-2001) are not discussed and are very relevant to this study.
- 4) The authors should also justify the choice of rhenium as a dopant given the low intrinsic stability based on the Pourbaix diagram.

Results and discussion:

- 1) Tafel slopes reported for commercial RuO₂ deviate from those reported in the literature, which are normally of ca. 60 mV dec⁻¹. (Energy Environ. Sci. 2022,15, 1977-1987)
- 2) Turnover frequency values are stated in h⁻¹ in this manuscript. This should be amended as these are normally reported in s⁻¹ (see JACS Au 2021, 1, 5, 586-597 for representative work in electrocatalysis devoted to TOF estimation)
- 3) OER long-term testing was performed on carbon paper, which is known not to be the best catalyst support for OER benchmarking due to substrate passivation (ChemSusChem 2017, 10, 4140-4143; J. Mater. Chem. A 2018, 6, 14162-14169; Catal. Today 2017, 295, 32-40). This, along with the catalyst ink/dropcasting/spraycoating quality (J. Power Sources 2017, 353 19-27; Electrochem. Commun. 2017, 85, 1-5) could be artifacts which lead to the low stability of RuO₂ and the much prolonged operation of the ReRuO₂ catalyst. The authors should discuss these points and their implications.
- 4) The authors monitor Re and Ru stability with ICP-MS measurements on the liquid samples. It is unclear whether the loss percentages reported refer to the total catalyst loading or if they are normalized based on the relative elemental content. This should be clarified. In addition, the authors should evaluate if and if there is any time-dependent stabilization of the catalyst based on the S-number metric as reported previously (Nat. Commun. 2021, 12, 2231). Finally, Re vacancies should be formed: can these be identified in HAADF-STEM after long-term testing?
- 5) HAADF-STEM imaging on RuO₂ after 20 h testing (Suppl. Fig. 16) still showcases clear lattice domains in the NPs. The authors state a “dissolution-induced lattice collapse”, which does not seem to apply here.
- 6) The very interesting dynamic oxidation state of Re L₃-edge XAS study should be backed by a side-by-side comparison of the Ru K-edge, which is surprisingly not reported here and should be the mirror image of the results on Re. This point is key as the proposed pathway change to AEM relies on that.
- 7) Reference 36 refers to an ATR-SEIRAS study performed on ORR. A more suitable reference should be found. In addition, the Raman feature at 1300 cm⁻¹ is fairly sharp in RuO₂ but faint in ReRuO₂ and should be discussed as it also presents a potential-dependent intensity.

8) The isotope-labelled OER experiments provide interesting information, but it would have made more sense to perform them at the studied OER potentials chosen for XAS so that a clear link between the dynamic role of Re and the reaction pathway would be revealed. If possible, I would recommend the authors to perform them accordingly.

9) Analogous to the cited work Zagalskaya et al. (ACS Catal. 2020, 10, 3650–3657), the manuscript would benefit from DFT calculation which would devote to the role of Ru or Re defects in the OER electrocatalytic pathway as ICP-MS data clearly shows preferential defect formation (see my previous comment)

Supplementary information:

1) Fig. 2: Include JPCD diffraction patterns of RuO₂ and ReO₂ should be included for direct comparison

2) Fig. 13: colours used in Re 4f peak deconvolution should be accompanied with a legend related to the specific oxidation state they are ascribed to

3) Table 1: two numbers are reported for the Ru_{0.06}Re_{0.94}O₂ mass activity. To what do they correspond to?

Response to Reviewer #1

Reviewer's Remarks to Authors

This manuscript reports Re doped RuO₂ (Re_{0.06}Ru_{0.94}O₂) catalyst that exhibits significantly improved activity and stability for acidic OER. The authors propose that during operation, the Re dopants in RuO₂ exhibits dynamic electron-accepting-donating behavior in which the Re gains electrons to activate the Ru site for a high OER activity and donates electrons back at large overpotential to maintain a high stability. The dynamic behaviour was followed mainly using operando Re L-edge X-ray absorption spectroscopy (XAS). Several other nice in-situ techniques such as Infrared absorption spectroscopy, differential electrochemical mass spectrometry (DEMS) were used as well, suggesting the OER mechanism for Re_{0.06}Ru_{0.94}O₂ is based on AEM, and the AEM renders the high stability of Re-RuO₂ catalyst in acidic conditions. Overall, the obtained results are interesting and this manuscript will be of broad interest and importance in Nat. Comm. However, there are a few issues to be clarified.

Response

We thank Reviewer #1 for his/her valuable comments and positive recommendation.

Comment 1-1

What is the spin configuration of Ru⁴⁺ in RuO₂? It is hard to understand the peak assignments for t_{2g} and e_g at O K edge XAS spectra shown in Supplementary Fig. 6 and Fig. 12.

Response

The 4d⁴ system RuO₂ has a rutile structure, in which the cation is in six-fold elongated octahedral coordination and the oxygen is threefold coordinated (*Chem. Rev.* 2020, 120, 4056–4110 and *Phys. Scr.* 1977, 16, 351).

The t_{2g} and e_g for our sample were assigned based on published reports (*Phys. Rev. B* 2000, 61, 5262 and *Appl. Phys. Lett.* 2007, 90, 042108). The two 'sharp' features denoted are attributable to the excitation of the O 1s core electrons into hybridized states between O 2p and Ru 4d t_{2g} and e_g states because of splitting by the crystal field. The sharp peak at ca. 528 eV refers to t_{2g} states and is followed by a broader peak at ca. 533 eV related to e_g states.

In response to this comment of Reviewer #1, we have in our R-SI, p. 5, caption of Supplementary Fig. 6, revised the text to read:

'The sharp peak at ca. 528 eV refers to t_{2g} states and is followed by a broader peak at ca. 533 eV related to e_g states. The two 'sharp' features denoted are attributable to the excitation of the O 1s core electrons into hybridized states between O 2p and Ru 4d t_{2g} and e_g states because of splitting by the crystal field.'

Comment 1-2

It is mentioned in page 7 that "The O related spectra for RuO₂ evidence an electron transfer from e_g to t_{2g} caused by Ru dissolution induced lattice collapse and catalyst degradation". It should be noted that the peaks at O K-edge spectra represent the unoccupied state of transition metal cation. How can one evidence from the O XPS and K-edge XAS spectra that there is electron transfer from e_g to t_{2g} ?

Response

We agree with Reviewer #1 that we should evidence there is electron transfer from e_g to t_{2g} . O XPS and K edge spectra confirm a charge redistribution instead of electron transfer from e_g to t_{2g} .

In response to address this comment directly, we have in our R-MS, p. 8, para. 1, revised the text to read:

'O-related spectra for RuO₂ evidence higher average Ru valence states and charge redistribution compared with pristine sample, that is caused by Ru dissolution-induced catalyst degradation (Supplementary Figs. 17c, d)^{48,49}.

Comment 1-3

In page 7, "Further EXAFS fitting showed that the Ru-O peak ...(the coordination number N is 1.8)...". Such a low coordination number of 1.8 seems not possible.

Response

The low coordination number of 1.8 corresponds to a sub-shell for Ru-O coordination. Actually, from the Ru K-edge EXAFS fitted results (**Supplementary Fig. 18** and **Supplementary Table 3**), it can be seen that the Ru-O peak can be divided into two distinct sub-shells with interatomic distances of 1.92 Å (coordination number N is 1.8) for Ru-O1 and 2.01 Å (N is 4.1) for Ru-O2, similar to the pristine sample. This Ru-O bond-length distribution corresponds to a total Ru-O coordination number of 5.9 and, therefore, evidences the adoption of distorted Ru-O₆ octahedra. Identical Ru-O bond distance distribution for the RuO₆ octahedra is shown in the standard RuO₂ crystal with rutile structure.

Comment 1-4

The EXAFS data fittings at Ru K edges (Supplementary Fig. 18), Re L edge (Supplementary Fig. 19 and Supplementary Fig. 20) do not look fitted well. More careful analysis and re-fitting the EXAFS data should be carried out. The description of the fitting procedure and the approximations used should be given in the Supplemental Materials.

Response

We agree with Reviewer #1. All EXAFS data at Ru K-edge and Re L₃-edge under *ex-situ* and *operando* conditions were carefully analyzed, re-fitted, and updated. As is seen from **Figures R2-4** and **Tables R1-2**, the quality of the EXAFS fitting has been significantly improved.

Figure R2. Ru K-edge EXAFS fitted analyzes for $\text{Re}_{0.06}\text{Ru}_{0.94}\text{O}_2$ prior to and following 50 h stability test. Best-fit parameters are given in **Supplementary Table 3**.

In response to address directly this comment we have in our:

1) R-SI;

i) Added **Figs. R2-R4** as **Supplementary Figs. 18, 20-21**

ii) Included **Tables R1-R2** as **Supplementary Tables 3-4**

2) R-MS, p. 23, para. 2, included additional text for description of the fitting-procedure and the approximations used in the *Methods* section as follows;

'XAS analysis was carried out with standard procedures using ATHENA and ARTEMIS modules implemented in IFEFFIT software⁵⁷. The EXAFS signal was obtained via background subtraction and normalization, and $\chi(k)$ data Fourier-transformed to real (R) space using a Hanning window. Least-squares curve-fitting of EXAFS $\chi(k)$ data was carried out in R-space to determine quantitative structural parameters around central atoms. Theoretical structural models were

constructed on the basis of crystal structures for RuO_2 and Re-RuO_2 , with scattering amplitudes, phase shifts and photoelectron mean free path for all paths computed with *ab initio* code FEFF 8.6.'

Figure R3. Re L_3 -edge EXAFS fitted analyzes for $\text{Re}_{0.06}\text{Ru}_{0.94}\text{O}_2$ prior to and following 50 h stability test. Best-fit parameters are given in **Supplementary Table 4**

Figure R4. *Operando* Re L_3 -edge EXAFS fitted analyzes for $\text{Re}_{0.06}\text{Ru}_{0.94}\text{O}_2$ at differing applied potentials. Measured and computed spectra are in good agreement. Best-fit parameters are given in **Supplementary Table 4**.

Table R1. Ru K-edge EXAFS fitted parameters for $\text{Re}_{0.06}\text{Ru}_{0.94}\text{O}_2$ ^a

Sample	Shell	N	R (Å)	σ^2 (Å ²)	ΔE_0 (eV)	R_f %
Re_{0.06}Ru_{0.94}O₂	Ru-O1	1.8	1.92	0.003	6.5	0.2
Pristine^b	Ru-O2	4.2	2.01	0.004		
Re_{0.06}Ru_{0.94}O₂	Ru-O1	1.8	1.92	0.003	6.6	0.2
after OER^b	Ru-O2	4.1	2.01	0.004		
Re_{0.06}Ru_{0.94}O₂	Ru-O1	1.8	1.94	0.003	6.1	0.2
OCP^b	Ru-O2	3.9	2.00	0.004		
Re_{0.06}Ru_{0.94}O₂	Ru-O1	1.8	1.93	0.003	8.5	0.1
1.2 V^b	Ru-O2	3.8	2.00	0.004		
Re_{0.06}Ru_{0.94}O₂	Ru-O1	1.8	1.92	0.003	8.7	0.1
1.3 V^b	Ru-O2	3.8	2.00	0.004		
Re_{0.06}Ru_{0.94}O₂	Ru-O1	1.8	1.92	0.003	8.5	0.2
1.4 V^b	Ru-O2	3.8	2.00	0.004		
Re_{0.06}Ru_{0.94}O₂	Ru-O1	1.8	1.91	0.003	8.5	0.1
1.5 V^b	Ru-O2	4.0	2.01	0.004		
Re_{0.06}Ru_{0.94}O₂	Ru-O1	1.8	1.91	0.003	9.0	0.1
1.6 V^b	Ru-O2	3.9	2.01	0.004		

^a N , coordination number, R , distance between absorber and backscatter atoms, σ^2 , Debye–Waller factor to account for both thermal and structural disorders, ΔE_0 , inner potential correction, R_f factor (%) for goodness of fit. Error bounds (accuracies) that characterize structural parameters obtained *via* EXAFS spectroscopy were estimated as, $N \pm 20\%$, $R \pm 1\%$, $\sigma^2 \pm 20\%$ and $\Delta E_0 \pm 20\%$. S_0^2 was fixed at 1.0 as determined from RuO_2 reference fitting. Ru-O1 and Ru-O2 represent the first and second nearest neighbour coordination O atoms. Bold numbers show fixed coordination number (N) according to crystal structure. ^b Fitted range, $2.5 \leq k$ (Å⁻¹) ≤ 10.7 and $0.8 \leq R$ (Å) ≤ 2.3 .

Table R2. Re L₃-edge EXAFS fitted parameters for Re_{0.06}Ru_{0.94}O₂.^a

Sample	Shell	N	R (Å)	σ^2 (Å ²)	ΔE_0 (eV)	R_f %
Re_{0.06}Ru_{0.94}O₂	Re-O1	3.8	1.82	0.004	7.8	0.6
Pristine^b	Re-O2	1.9	1.93	0.005		
Re_{0.06}Ru_{0.94}O₂	Re-O1	3.9	1.81	0.004	6.7	0.3
after OER^b	Re-O2	1.8	2.00	0.005		
Re_{0.06}Ru_{0.94}O₂	Re-O1	3.7	1.82	0.004	8.2	0.1
OCP^b	Re-O2	2.0	2.01	0.005		
Re_{0.06}Ru_{0.94}O₂	Re-O1	4.1	1.82	0.004	7.9	0.7
1.2 V^b	Re-O2	1.9	2.03	0.005		
Re_{0.06}Ru_{0.94}O₂	Re-O1	3.4	1.80	0.004	6.8	0.6
1.3 V^b	Re-O2	1.9	1.97	0.005		
Re_{0.06}Ru_{0.94}O₂	Re-O1	3.9	1.79	0.004	6.3	0.6
1.4 V^b	Re-O2	1.9	1.98	0.005		
Re_{0.06}Ru_{0.94}O₂	Re-O1	4.7	1.81	0.004	6.8	0.3
1.5 V^b	Re-O2	1.2	2.06	0.005		
Re_{0.06}Ru_{0.94}O₂	Re-O1	3.6	1.81	0.004	7.5	0.4
1.6 V^b	Re-O2	1.9	1.96	0.005		

^a N , coordination number, R , distance between absorber and backscatter atoms, σ^2 , Debye–Waller factor to account for both thermal and structural disorders, ΔE_0 , inner potential correction, R_f factor (%) for goodness of the fit. Error bounds (accuracies) that characterize the structural parameters obtained *via* EXAFS spectroscopy were estimated as, $N \pm 20\%$, $R \pm 1\%$, $\sigma^2 \pm 20\%$ and $\Delta E_0 \pm 20\%$. S_0^2 was fixed at 0.75 as determined from Re⁷⁺ aq. reference fitting. Re-O1 and Re-O2 represent the first and second nearest neighbour coordination O atoms. Bold numbers show fixed coordination number (N) according to crystal structure. ^b Fitted range, $1.8 \leq k$ (Å⁻¹) ≤ 9.0 and $0.8 \leq R$ (Å) ≤ 2.3 .

Comment 1-5

The authors only show the dynamic changes of oxidization state and bonding length for Re cations using operando XAS. How do the Ru cations change accompanying the dynamic behavior of Re?

Response

To address this comment, we carried out *operando* Ru K-edge XAS at differing overpotentials to understand the change for Ru cations, **Figure R5**. In contrast to dynamic change for Re cations with applied potentials, it was observed that both the *operando* Ru K-edge XANES and EXAFS exhibited only ‘slight’ change during catalysis, which was confirmed by detailed Ru K-edge EXAFS fitted analyses, **Figures R6, 7** and **Table R1**. It is widely acknowledged that the XAS analyses reflect average information for all Ru atoms in $\text{Re}_{0.06}\text{Ru}_{0.94}\text{O}_2$. The XAS signal for Ru-O-Ru sites dominates and overlaps with that for Ru-O-Re sites because of the low doping amount of Re in $\text{Re}_{0.06}\text{Ru}_{0.94}\text{O}_2$, leading to the unchanged Ru K-edge spectra. Therefore, we focused mainly on the Re L₃-edge to determine dynamic behaviour for Re-O-Ru sites. In addition, the unchanged Ru K-edge spectra evidence boosted stability for $\text{Re}_{0.06}\text{Ru}_{0.94}\text{O}_2$ *via* Re doping.

Figure R5. a, Ru K-edge XANES spectra for $\text{Re}_{0.06}\text{Ru}_{0.94}\text{O}_2$ at differing potential in O_2 -saturated 0.1 M HClO_4 . **b**, FT-EXAFS signals for $\text{Re}_{0.06}\text{Ru}_{0.94}\text{O}_2$ corresponding to **a**.

In response to fully address this comment of Reviewer #1 we have in our:

- 1) R-SI, added **Figs. R5-R7** as **Supplementary Figs. 22-24**
- 2) R-MS, p. 13, para. 3, included explanatory text, namely;

We investigated the Ru sites via operando Ru K-edge XAS to determine the change for Ru cations. In contrast to dynamic change for Re cations with applied potential, it was found that both operando Ru K-edge XANES and EXAFS exhibit only ‘slight’ change during catalysis (Supplementary Fig. 22), as was confirmed by detailed Ru K-edge EXAFS fitted analyses (Supplementary Figs. 23-24 and Supplementary Table 3). It is widely acknowledged that XAS

analyses reflect average information for all Ru atoms in $\text{Re}_{0.06}\text{Ru}_{0.94}\text{O}_2$. The XAS signal for Ru-O-Ru site dominates and overlaps that for Ru-O-Re site because of the low doping amount of Re in $\text{Re}_{0.06}\text{Ru}_{0.94}\text{O}_2$, leading to the unchanged Ru K-edge spectra. Therefore, we focused mainly on the Re L₃-edge to determine the dynamic behaviour of Re-O-Ru sites.'

Figure R6. Operando Ru K-edge EXAFS fitted analyzes for $\text{Re}_{0.06}\text{Ru}_{0.94}\text{O}_2$ at differing applied potential. Measured and computed spectra are in good agreement. Best-fit parameters are given in **Supplementary Table 3**.

Figure R7. Change in bond length for Ru–O1 and Ru–O2 coordination shells.

Comment 1-6

It is hard to see the difference in the Re L-edge from Figure 3a.

Response

In response to address this, we have, in our R-MS, revised **Figure 3a** to highlight the change in the Re L₃-edge more clearly.

Fig. R8. Re L₃-edge XANES spectra for Re_{0.06}Ru_{0.94}O₂ at differing potential in O₂-saturated 0.1 M HClO₄. Right part is the contour plot for the Re L₃-edge white peak intensity.

Response to Reviewer #2

Reviewer's Remarks to Authors

In this work, the authors reported the Re dopants in RuO₂ that can undergo a dynamic electron accepting-donating, in this manner, the static electron redistribution induced by the conventional dopants can be well mitigated. Accordingly, the Re_{0.06}Ru_{0.94}O₂ was reported to show highly desired electrocatalytic performance with high mass activity and high stability. It is a piece of interesting work to develop viable electrocatalysts toward acidic OER, thus it can be accepted for publication in Nat Commun after suitable revision:

Response

We thank Reviewer #2 for his/her valuable comments and positive recommendation for publication.

Comment 2-1

The sample Re_{0.06}Ru_{0.94}O₂ has a relatively low Re doping, how did the authors determine the atomic ratio between Re/Ru, in addition, is it possible to tune the ratio by adjusting the synthetic condition, and one may wonder if the synthesis of such Re_{0.06}Ru_{0.94}O₂ (with fixed ratio) is repeatable, moreover, it is interesting to study if the ratio will affect the electrocatalytic properties. Given the ratio of Re/Ru indeed varied during testing, will the variation of atomic ratio of Re/Ru degrade or promote the electrocatalytic performance?

Response

The Re/Ru ratio was determined *via* inductively coupled plasma mass spectrometry (ICP-MS). Because the ICP-MS has a very high accuracy of up to 0.01 ppb, Re/Ru ratio can be precisely determined. The Re atoms were doped in RuO₂ lattice *via* the following:

Excess NaNO₃ ensures 'complete' oxidation of RuCl₃ to RuO₂. Therefore, the Re doping level is controlled by changing the amount of NaReO₄ in the molten salt. Because the only variation during synthesis is the amount of NaReO₄, Re_{0.06}Ru_{0.94}O₂ and other samples with fixed ratios are repeatable. Additionally, we successfully obtained Re-RuO₂ with different doping ratios, **Figure R9**. Re doping does not change the rutile structure of RuO₂.

In response to address this comment of Reviewer #3 fully, we have in our:

1) R-SI, added **Figure R9** as **Supplementary Fig. 9**

2) R-MS, p.5, para. 1, included the following clarifying text;

*‘The doping impact of Re on OER performance was determined (**Supplementary Figs. 9-14**) via a series of Re-RuO₂ with different Re. It was found that Re doping does not measurably change rutile structure for Re-RuO₂.’*

Figure R9. a, Re and Ru ratio in samples with change of Re and Ru ratio in precursors. **b**, XRD pattern for Re-RuO₂ samples with differing Re.

Additionally, we investigated the impact of amount of Re in Re-RuO₂ on electrocatalytic performance. **Figure R10** presents the relationship between activity and Re in Re-RuO₂ as a volcano plot. Re_{0.06}Ru_{0.94}O₂ exhibited ‘best’ performance with the lowest η_{10} and Tafel slope. The over-doping induced activity decay is because Re dopants are inert for OER, based on DFT computations. Therefore, high doping levels lower the density of the surface active site and decrease OER. In addition, we computed the Re and Ru change during OER. Following 200 h OER test, just 0.34 % Ru and 0.62 % Re were dissolved in the electrolytes. Therefore, the difference in Ru/Re ratio during OER is negligible.

We have, therefore, in our R-SI:

3) Added **Figure R10** as **Supplementary Fig. 14**

4) p. 9, included additional text as follows;

*‘As is presented in **Supplementary Fig. 14**, the relationship between activity and Re doping level in Re-RuO₂ is a volcano plot. Re_{0.06}Ru_{0.94}O₂ exhibited the ‘best’ performance with the lowest η_{10} and Tafel slope. This is because Re dopants are inert for OER, compared to Ru site. High doping levels lower the density of the active site on the surface and, decrease overall performance.’*

Figure R10. **a**, LSV curves for Re-RuO₂ with differing Re in O₂-saturated 0.1 M HClO₄. **b**, η_{10} for Re-RuO₂ electrocatalyst. **c**, Tafel plot for Re-RuO₂ with differing Re corresponding to **a**.

Comment 2-2

The author will be appreciated if they can discuss the definition or difference regarding static electron redistribution and dynamic electron accepting-donating in the introduction section;

Response

We agree with Reviewer #3 to discuss the definition of static electron and dynamic electron accepting-donating in the introduction.

In response, we have in our R-MS, p. 3, para. 2, included the following additional discussion text:

‘For example, even though conventional heteroatom doping strengthens the lattice oxygen in RuO₂ at small overpotential via electronic structural redistribution, the stability is not sufficient for practical application because of demetallation of the modified-Ru site at large overpotentials³⁶. Therefore, dopants that can tune OER performance via dynamic electron distribution under differing potentials are practically attractive.’

Comment 2-3

The LSV in Fig. 1a is obviously obtained by i-R compensations, the authors should detail the parameter of i-R compensations, and the LSV curves without i-R compensations are suggested to be supplied at least in the supporting information section so that the readers can see the practical status;

Response

Electrochemical data were corrected for uncompensated series resistance R_s which was determined through open-circuit voltage. A 90 % i-R compensation was selected to determine R_s . The final

polarization curve was determined *via* *i*-R compensation. The uncompensated LSV curves determined in 0.1 M HClO₄ are shown in **Figure R11**.

In response to directly address this comment of Reviewer #3 we have in our R-MS:

1) p. 5, para. 1, included the following text;

*'LSV curves without *i*-R compensation are presented in **Supplementary Fig. 7a** as a reference.'*

2) p. 21, para. 1, included text, namely;

*'Electrochemical data were corrected for uncompensated series resistance R_s that was determined via open-circuit voltage. A 90 % *i*-R compensation was selected to determine R_s . The final polarization curve was determined via *i*-R compensation.'*

Figure R11. LSV curves for catalysts without *i*-R compensation.

Comment 2-4

The doped Re atoms in Re_{0.06}Ru_{0.94}O₂ are highlighted by yellow circle in its corresponding Aberration corrected HAADF-STEM image (Fig. 2b), however, it is confusing why the authors can verify these dots are Re rather than Ru;

Response

In HAADF-STEM heavy atoms are brighter than light atoms. The formation of distinctive bright and dark spots is interpreted as the difference in the degree of localization and inelastic absorption of channeling electrons in individual atoms by analyses of convergent wave fields inside the crystal in both real and reciprocal space (*Ultramicroscopy*, 2012, 120, 48.).

In response, because the atomic weight of Re (186.21) is significantly greater than that for Ru (101.07), the bright dots in **Fig. 2b** can be verified as Re atoms.

Therefore no change has been made to our R-MS to this comment of Reviewer #2.

Comment 2-5

In this work, the activity and stability were evaluated by a different way from the previous paper, i.e., the mass activity and stability number of S-number, the area activity should be supplied as this parameter is more meaningful for performance evaluation, and please detail the description on how to calculate stability number of S-number;

Response

The area activity was computed by normalizing the electrochemical surface area of different catalysts, **Figure R12**.

The specific area activity for $\text{Re}_{0.06}\text{Ru}_{0.94}\text{O}_2$ outperformed the other samples, demonstrating superior intrinsic OER catalytic activity.

In response to address fully this comment of Reviewer #2, computational details for normalized area activity and S-number have now been included in our R-SI, namely:

1) In our R-SI, p. 29;

‘Supplementary Note 4

Specific area activity

The specific area activity was determined by normalizing the ECSA for different catalysts. The specific current density per ECSA (j_a) was computed from:

$$j_a = \frac{j_{geo} \times A_{geo} \times C_s}{C_{dl}} \quad (S7)$$

*where j_{geo} is the geometric area current density and A_{geo} the geometric area of the glassy carbon electrode (0.19625 cm^{-2}). C_{dl} was measured from CV in **Supplementary Fig. 11**. The C_s for 0.035 mF cm^{-2} was used to estimate ECSA.’*

2) R-SI, p. 30;

‘Supplementary Note 6

Stability number

The stability number for catalysts (S-number) was computed from:

$$\text{Stability number} = \frac{N_{\text{oxygen}}}{N_{\text{noble metal}}} \quad (S9)$$

where N_{oxygen} is the molar number of total oxygen evolved within a period of time (computed from total charge). The total charge is obtained from $t \times I$. Because the Faradaic efficiency is 100 %, the molar number for O_2 in 200 h is 0.0187 mol. The $N_{\text{noble metal}}$ total dissolved noble metal number was measured via ICP-MS.'

Figure R12. LSV curves for catalysts normalized to ECSA.

Response to Reviewer #3

Reviewer's Remarks to Authors

The manuscript by Jin et al. devotes to the study of ruthenium oxide dynamic doping by incorporating rhenium into the catalyst matrix. The work presents a large set of characterization techniques to support the claims that Re plays the role of a dynamic electron donor-acceptor, additionally promoting the adsorbate evolution mechanism. I believe that the work merits to be published in Nature Communications, although some aspects of the manuscript need to be improved.

Response

We thank Reviewer #3 for his/her valuable comments and positive recommendation for publication.

Comment 3-1

Introduction: The authors aimed to summarize very important works in a very short section, and by doing so they fail to cite key literature in the field. See examples as follows:

Reference 8 is not the first report of RuO₄ detection: should be substituted by J. Electroanal. Chem. 1987, 237, 251-260?

Response

We agree with this comment of Reviewer #3.

In response to address this directly, we have in our R-MS:

1) Cited this as Reference 8.

2) In R-MS, p. 3, para. 1, included additional text as follows;

'In 1987 researchers reported that rutile-phase ruthenium oxide (RuO₂) has significant practical potential to replace IrO₂ for acidic OER because of excellent activity⁸.'

Comment 3-2

The RuO₂ hybridization with IrO₂ has been discussed in more relevant works: J. Electrochem. Soc. 2016, 163, F3099; ACS Appl. Energy Mater. 2020, 3, 5229-5237; ACS Catal. 2021, 11, 15, 9300-9316; Electrochim. Acta 2012, 70, 158-164; Appl. Catal. B: Environ. 2015, 164, 488-495.;

Response

In response to address fully this comment, we have in our R-MS:

1) Cited these papers

2) R-MS, p. 3, para. 1, included text as follows;

'In recent years, research has focused on boosting structural stability of RuO₂ or Ru-based electrocatalysts via hybridization with IrO₂¹⁷⁻²⁴.'

Comment 3-3

Besides that, works by Chorkendorff and co-workers (Chem. Sci., 2015, 6, 190; Energy Environ. Sci., 2022, 15, 1988-2001) are not discussed and are very relevant to this study.

Response

We agree with this comment of Reviewer #3.

In response to address this comment, we have in our R-MS:

1) Included these papers

2) Added clarifying text, namely;

i) p. 3, para. 1

'However, achieving high activity and stability with this catalyst is practically difficult because of the instability of the surface Ru sites in RuO₂⁹⁻¹¹.'

ii) p. 3, para. 2

'For example, it was reported that stability and activity of RuO₂ nanoparticles can be boosted by reinforcing Ru-O bonding via surface heat treatment¹⁰.'

iii) p. 16, para. 2

'Importantly, it was reported that < 0.2 % of evolved oxygen contains an oxygen atom originating from RuO_x¹¹. The difference with our findings is because of poor crystallinity of our sample obtained from the molten salt method, which has more active lattice oxygen.'

Comment 3-4

The authors should also justify the choice of rhenium as a dopant given the low intrinsic stability based on the Pourbaix diagram.

Response

Reported findings have demonstrated incorporation of high-valence metals, including W and Ta to stabilize otherwise unstable low-charge Ir/Ru. However, no reported study has been made on Re, an element with multiple oxidation states between -3 and +7. In addition, rhenium oxides are acidic oxides with strong metal-oxygen bonds. Therefore, Re is highly suitable as a dynamic dopant that can stabilize the lattice oxygen and low-charge metal active sites.

In response to address this comment of Reviewer #3 directly, we have in our R-MS, p. 4, para. 1 included text justifying the choice of rhenium, namely:

'Reported findings have demonstrated incorporation of high-valence metals, such as W and Ta to stabilize low-charge Ir/Ru^{3+,37}. Doping with Re, an element with multiple oxidation states between -3 and +7, into RuO₂, it is possible to build adaptive active sites with dynamic electron transfer. Additionally, rhenium oxide is an acidic oxide with strong metal-oxygen bonds. Therefore, Re dopants stabilize the lattice oxygen and low-charge metal active sites concurrently.'

Comment 3-5

Results and discussion:

Tafel slopes reported for commercial RuO₂ deviate from those reported in the literature, which are normally of ca. 60 mV dec⁻¹. (Energy Environ. Sci. 2022,15, 1977-1987).

Response

The commercial ruthenium (IV) oxide ($\geq 99.9\%$) was purchased from Sigma-Aldrich without further purification. All LSV curves reported in our MS were determined following 10 CV scans to remove surface-adsorbed contaminations and stabilize the samples. In the article, *Energy Environ. Sci.* 2022,15, 1977-1987 and, cited references, nanoparticle samples are synthesized *via* the researcher. In our work, the synthesized RuO₂ nanoparticle exhibited a Tafel slope of 50.3 mV dec⁻¹, which is close to the *ca.* 60 mV dec⁻¹. The commercial RuO₂ in our work exhibited 76.4 mV dec⁻¹ because of poor stability. **Figure R13** presents Tafel slope for C-RuO₂ determined from the first scan, as 63.7 mV dec⁻¹, close to *ca.* 60 mV dec⁻¹. However, during the CV scans, the catalyst exhibited decay because of poor stability.

Figure R13. **a**, LSV curves for C-RuO₂ prior to and following 5 CV scans. **b**, Tafel plot for C-RuO₂ prior to and following 10 CV scans.

In response to address this comment of Reviewer #3 directly, we have in our R-MS, p. 21, para. 1, included a revised text, namely:

‘To evaluate OER for different catalysts, the working electrode was put through 10 cyclic voltammetry (CV) scans between 1.1 to 1.6, V at a 20 mV s^{-1} to clean and stabilize the surface of the catalyst.’

Comment 3-6

Turnover frequency values are stated in h^{-1} in this manuscript. This should be amended as these are normally reported in s^{-1} (see JACS Au 2021, 1, 5, 586–597 for representative work in electrocatalysis devoted to TOF estimation).

Response

We agree with this comment of Reviewer #3.

In response to address this fully we have in our R-MS:

- 1) R-MS, changed to s^{-1} , see **Figure R14**
- 2) R-SI, included this figure as **Supplementary Information Fig. 8a**
- 3) R-MS, p. 5, para. 1, included the additional clarifying text;

‘The turnover frequency (TOF)³⁹ for $\text{Re}_{0.06}\text{Ru}_{0.94}\text{O}_2$ at overpotential 272 mV is 0.17 s^{-1} (Supplementary Fig. 8a), which is an order of magnitude greater than for C-RuO₂ of 0.004 s^{-1} .’

Figure R14. TOF for $\text{Re}_{0.06}\text{Ru}_{0.94}\text{O}_2$, RuO_2 and C-RuO₂.

Comment 3-7

OER long-term testing was performed on carbon paper, which is known not to be the best catalyst support for OER benchmarking due to substrate passivation (ChemSusChem 2017, 10, 4140-4143; J. Mater. Chem. A 2018, 6, 14162-14169; Catal. Today 2017, 295, 32-40). This, along with the catalyst ink/dropcasting/spraycoating quality (J. Power Sources 2017, 353 19-27; Electrochem. Commun. 2017, 85, 1-5) could be artifacts which lead to the low stability of RuO₂ and the much prolonged operation of the ReRuO₂ catalyst. The authors should discuss these points and their implications.

Response

We agree with Reviewer #3 that carbon paper is not the best support for OER testing because of passivation at large positive potential. Therefore, we conducted multiple experiments to exclude the impact of carbon paper.

The stability tests for Re_{0.06}Ru_{0.94}O₂ and RuO₂ were conducted at similar overpotential of < 1.5 V. Based on our results and published findings (e.g. *Nat. Catal.* 4, 2021, 1012-1023 and *Nat. Commun.* 2022, 13, 4871) the carbon paper electrode is suitable for OER testing under these overpotentials. In addition, based on *ChemSusChem* 2017, 10, 4140-4143 we determined stability of our materials *via* monitoring Ru dissolution in the electrolyte using ICP-MS, **Fig. 1e**. The RuO₂ has a linear-dependent increased Ru dissolution that evidences a fast decay of catalyst. However, Ru dissolution in Re_{0.06}Ru_{0.94}O₂ is slower. We can conclude, therefore, that Re_{0.06}Ru_{0.94}O₂ has better OER stability than RuO₂.

In response to directly address this comment, we have in our R-MS:

1) p. 6, para. 1, included additional explanatory text, namely:

‘Importantly, carbon paper is not an ideal support for acidic OER durability testing because of due substrate passivation⁴²⁻⁴⁶. As a result, chronopotentiometry is not always a reliable technique to determine stability of acidic OER catalysts on carbon paper. Detecting catalyst mass losses during OER can provide quantitative information that distinguishes between different degradation mechanisms.⁴² Therefore, Ru dissolution in different catalysts was determined to confirm a degradation mechanism.’

We also agree with Reviewer #3 that the quality of catalyst ink drop casting/spray-coating influences stability testing. Therefore, we used the same method to prepare the electrode of RuO₂ and Re_{0.06}Ru_{0.94}O₂ to exclude the impact of coating method on stability testing. Stability

measurements were carried out *via* uniform air-brush spraying of the catalyst on the carbon with a loading mass of $0.1 \text{ mg}_{\text{cata}} \text{ cm}^{-2}$. The stability test was conducted at temperature of $25 \text{ }^\circ\text{C}$. This method is recently reported, e.g. *Nat. Commun.* 2022, 13, 4871 and *J. Am. Chem. Soc.* 2021, 143, 6482–6490.

Therefore additionally, we have in our R-MS:

2) p. 21, para. 1, included additional detailed discussion in **Method** as follows;

‘To determine metal dissolution in the electrolyte, a stability test was conducted in an H-cell. The stability measurements were carried out by air-brush spraying the catalysts uniformly on the carbon with a loading mass of $0.2 \text{ mg}_{\text{cata}} \text{ cm}^{-2}$. The stability test was conducted under $25 \text{ }^\circ\text{C}$.’

Comment 3-8

*The authors monitor Re and Ru stability with ICP-MS measurements on the liquid samples. It is unclear whether the loss percentages reported refer to the total catalyst loading or if they are normalized based on the relative elemental content. This should be clarified. In addition, the authors should evaluate if and if there is any time-dependent stabilization of the catalyst based on the S-number metric as reported previously (*Nat. Commun.* 2021, 12, 2231). Finally, Re vacancies should be formed: can these be identified in HAADF-STEM after long-term testing?*

Response

The loss percentage for different catalysts is normalized based on the relative elemental content of Ru or Re. For example for RuO_2 , the loading mass is 0.2 mg ($0.2 \text{ mg cm}^{-2} \times 1 \text{ cm}^{-2}$). The Ru loading mass is 0.152 mg . The dissolved Ru concentration in the electrolyte is 104 ppb and the electrolyte volume is 40 mL . Therefore, the loss percentage for Ru is $(104 \text{ ppb} \times 40 \text{ mL})/0.152 \text{ mg} \times 100 \% = 2.7 \%$.

In response to address directly this comment of Reviewer #3 we have in our:

1) R-SI, p. 44, included the following additional text for computational details for ICP-MS data;

‘Supplementary Note 5

Loss computation

The loss of catalyst during stability testing was computed from:

$$\text{Loss percentage (\%)} = \frac{\text{Dissolved metal concentration} \times \text{Electrolyte volume}}{\text{Mass of metal in catalysts}} \times 100\% \quad (\text{S8})$$

where Dissolved metal concentration is obtained via ICP-MS and electrolyte volume is 40 mL . The Mass of metal in catalysts is computed based on ratio of metal in the catalyst. For example

for RuO₂, the loading catalyst on working electrode (1 cm⁻²) is 0.2 mg. The Ru loading mass is 0.2 × 0.76 = 0.152 mg.’

Our catalyst exhibited a time-dependent S-number, seen in **Supplementary Fig. 15**. The S-number for the catalysts increases with time and is attributed to a slower Ru dissolution rate.

To address this, we have in our;

2) R-MS, p. 7, para. 1, revised the text as follows

*‘The stability number (S-number) for the catalysts was determined via measuring oxygen produced and dissolved metal ion concentration in the electrolyte (**Supplementary Fig. 15**)⁴⁰. Both RuO₂ and Re_{0.06}Ru_{0.94}O₂ exhibited a time-dependent S-number, similar to recently reported findings⁴⁷. The S-number for the catalysts increases with time, and is attributed to slower Ru dissolution rate.’*

We agree with Reviewer #3 that Re and Ru vacancies may be formed following OER. However, Re vacancies are difficult to observe *via* HAADF-TEM because of the poor contrast between under-vacancy Ru atoms and, other Ru atoms. Therefore, we conducted XAS measurements for the samples prior to and following the OER stability tests, **Supplementary Table 3 and 4**. The Ru and Re coordination numbers did not apparently change, evidencing a low concentration of metal vacancies.

To address this, we have in our;

3) R-MS, p. 10, para. 1, included additional discussion text, namely

‘Additionally, the apparently unchanged coordination number excludes the formation of highly concentrated Ru/Re vacancies. Therefore the coordination environment for Ru in Re_{0.06}Ru_{0.94}O₂ following OER is confirmed to be unchanged, evidencing that Re doping strengthens the Ru–O bond and prevents Ru dissolution.’

Comment 3-9

HAADF-STEM imaging on RuO₂ after 20 h testing (Suppl. Fig. 16) still showcases clear lattice domains in the NPs. The authors state a “dissolution-induced lattice collapse”, which does not seem to apply here.

Response

The dissolution of metal species and the instability of the oxygen anion during OER causes the activity decay of RuO₂. As is shown in our work and in *Chem. Sci.*, 2015,6, 190-196, RuO₂ has much better stability than Ru nanoparticles, even with Ru dissolution. Therefore, it is reasonable

that the RuO₂ following OER testing retains a clear lattice, in accordance with the data in *Chem. Sci.*, 2015,6, 190-196.

In response to address this comment, we want to improve our description of ‘dissolution-induced lattice collapse’ and make it more accurate and have in our R-MS, p. 3, para. 1, included the following text:

‘Ru active sites are damaged in two ways, 1) formation of lattice-oxygen vacancies via lattice-oxygen-mediated mechanism (LOM)¹²⁻¹⁴, which leads to instability of the oxygen anion¹⁵, and 2) over-oxidation of Ru atoms to soluble RuO₄ species at high overpotential that leads to demetallation of the active sites¹⁶.’

Comment 3-10

The very interesting dynamic oxidation state of Re L₃-edge XAS study should be backed by a side-by-side comparison of the Ru K-edge, which is surprisingly not reported here and should be the mirror image of the results on Re. This point is key as the proposed pathway change to AEM relies on that.

Response

Please see detailed response to **Comment 1-5**.

Comment 3-11

Reference 36 refers to an ATR-SEIRAS study performed on ORR. A more suitable reference should be found. In addition, the Raman feature at 1300 cm⁻¹ is fairly sharp in RuO₂ but faint in ReRuO₂ and should be discussed as it also presents a potential-dependent intensity.

Response

We agree with this two-part comment of Reviewer #3.

In response to address this comment, in our R-MS:

- i) An additional two references regarding *OOH detection during OER are cited, namely, 1) *Nat. Cata.* 2019, 2, 304–313 and 2) *Nat. Commun.* 2022, 13, 5448).

It should be noted that *OOH bands shift toward a lower wavelength direction in alkaline solution compared with those in acidic solution because the H atom in *OOH can form a hydrogen bond with O atom in OH⁻, resulting in *OOH moving to a lower wavenumber direction in IR spectra.

Therefore, the *OOH band position in *Nat. Commun.* 2022, 13, 5448 is 1158 cm⁻², lower than our value.

ii) **Figure R15** presents the ATR-SEIRAS spectra of the blank, Au-coated Si prism at different potentials. The potential-dependent broad peak *ca.* 1260-1300 cm^{-1} is attributed to the oxidation of the Au-coated Si prism. The ‘sharper’ signal for RuO_2 than that for $\text{Re}_{0.06}\text{Ru}_{0.94}\text{O}_2$ is due to the lower *OOH peak intensity of RuO_2 during OER.

Figure R15. *In situ* ATR-SEIRAS spectra for blank Au-coated Si prism recorded during multi-potential steps.

Comment 3-12

The isotope-labelled OER experiments provide interesting information, but it would have made more sense to perform them at the studied OER potentials chosen for XAS so that a clear link between the dynamic role of Re and the reaction pathway would be revealed. If possible, I would recommend the authors to perform them accordingly.

Response

The isotope-labelled OER experiments cannot be performed under the same conditions as for XAS. We applied constant current chronopotentiometric method *operando* XAS characterization, which differs from isotope-labelled online DEMS using cyclic voltammetry.

Additionally, DEMS is suitable for detecting dissolved gases and not evolved gases. When overpotential reaches 1.6 V (large overpotential), significant evolved O_2 bubbles generate on the electrode surface, leading to a poor DEMS signal, as shown in **Figure R16**. Therefore, the

maximum current density for DEMS is *ca.* 2 to 2.5, mA (lower than 1.4 V) to obviate formation of evolved bubbles of O₂.

Therefore, no change has been made in our R-MS in response to this comment.

Figure R16. DEMS signals for ³²O₂ from reaction products for Re_{0.06}Ru_{0.94}O₂ in H₂¹⁶O aqueous sulphuric acid electrolyte with CV from 1.1 to 1.6 V.

Comment 3-13

Analogous to the cited work Zagalskaya et al. (ACS Catal. 2020, 10, 3650–3657), the manuscript would benefit from DFT calculation which would devote to the role of Ru or Re defects in the OER electrocatalytic pathway as ICP-MS data clearly shows preferential defect formation (see my previous comment).

Response

Qualitative analyses of the OER mechanism for RuO₂ and Re-RuO₂ with metal vacancies were investigated *via* DFT computation. Importantly, the structure for Re-RuO₂ with a Re vacancy is the same as for RuO₂ with a Ru vacancy following stabilization. We considered Re (vac-RuO₂) and Ru (vac-Re-RuO₂) vacancies on Re-RuO₂ to determine the impact on electrocatalytic performance. As shown in **Figure R17**, the sample with metal vacancies exhibits a reaction energy of 0.98 eV (vac-Re-RuO₂) and 1.01 eV (vac-RuO₂) on AEM pathway, respectively greater than that of 0.79 eV for Re-RuO₂. Additionally, we compared the minimum required energy for AEM, LOM and OPM on the defects, **Figure R18**. The defect sample exhibits optimized OPM

thermodynamic energy, less than that for AEM on Re-RuO₂. Therefore, the advantage of Re-RuO₂ over pure RuO₂ or defect is demonstrated.

Figure R17. Free energy diagram for OER on vac-RuO₂ and vac-Re-RuO₂ at 1.23 V vs. RHE, showing the six (6) possible intermediates for (110) surfaces. Dashed lines indicate unstable – OOH precursor states as H-stabilized OO*.

Figure R18. Minimum activation energy for different reaction pathways for vac-RuO₂ and vac-Re-RuO₂.

In response to address this comment of Reviewer #3 directly, we have in our:

- 1) R-SI, added **Figs. R17** and **R18**, respectively, as **Supplementary Information Figs. 36** and **37**
- 2) R-MS, p. 18, para. 1, included discussion text as follows;

*‘Qualitative analyses of OER mechanism for RuO₂ and Re-RuO₂ with metal vacancies were assessed via DFT computation. Importantly, the structure for Re-RuO₂ with a Re vacancy is the same as for RuO₂ with a Ru vacancy following stabilization. We considered Re (vac-RuO₂) and Ru (vac-Re-RuO₂) vacancies therefore on Re-RuO₂ to determine the impact on electrocatalytic performance. As is seen in **Supplementary Figs. 36 and 37**, although the sample with metal defects exhibits optimized OPM thermodynamic energy, it is less than that for AEM on Re-RuO₂, confirming the advantage of Re-RuO₂ over RuO₂ or metal defects.’*

Comment 3-14

Supplementary information:

Fig. 2: Include JPCD diffraction patterns of RuO₂ and ReO₂ should be included for direct comparison.

Response

We agree with Reviewer #3. In response to address this, we have in our R-SI included the JPCD diffraction patterns for RuO₂ and ReO₂ in **Supplementary Fig. 2**.

Figure R19. XRD pattern for $\text{Re}_{0.06}\text{Ru}_{0.94}\text{O}_2$ and RuO_2 . Both these samples exhibit a rutile phase. No ReO_x peak was apparent.

Comment 3-15

Fig. 13: colours used in Re 4f peak deconvolution should be accompanied with a legend related to the specific oxidation state they are ascribed to.

Response

In response to address this comment, we have in our R-SI revised **Supplementary Fig. 13**.

Figure R20. Re 4f XPS spectra for Re-RuO₂ with differing Re.

Comment 3-16

Table 1: two numbers are reported for the Ru_{0.06}Ru_{0.94}O₂ mass activity. To what do they correspond to?

Response

In response to address this comment in our R-SI, the Re-Ru mass activity is defined in **Supplementary Table 1** as follows:

“Re-related Ru mass activity. Based on DFT computations, the unsaturated Ru connected to Re is the active site for OER. Therefore, the Re-related Ru mass activity is computed based on the Re-related Ru mass, and not total Ru mass in Ru_{0.06}Ru_{0.94}O₂.”

END OF RESPONSE TO REVIEWS

REVIEWERS' COMMENTS

Reviewer #1 (Remarks to the Author):

The authors have addressed the questions/comments I raised. I recommend acceptance for publication.

Reviewer #2 (Remarks to the Author):

The authors well addressed the issues raised by the reviewers, and the revision is now acceptable for publication in Nature Communications as it is.

Reviewer #3 (Remarks to the Author):

The authors showcased an effort to tackle all the questions raised by myself and the other reviewers, which is appreciated. I believe that the manuscript can now be accepted in its current state.